# Walking in China's Historical and Cultural Streets: The Factors Affecting Pedestrian Walking Behavior and Walking Experience

Mimi Tian [1], Zhixing Li [1], Qinan Xia [1], Yu Peng [1], Tianlong Cao [1], Tianmei Du [1] and Zeyu Xing [2,*]

1   School of Design and Architecture, Zhejiang University of Technology, Hangzhou 310023, China
2   School of Management, Zhejiang University of Technology, Hangzhou 310023, China
*   Correspondence: xingzeyusmile@zjut.edu.cn; Tel.: +86-19103522318

**Abstract:** The urban street has evolved into an important indicator reflecting citizens' living standard today, and pedestrian walking activity in the streets has been proved to be a major facilitator of public health. Uncertainties, however, exist in the factors affecting pedestrian walking behavior and walking experience in streets. Especially, the factors affecting pedestrian walking behavior and walking experience in the historical and cultural streets. For the study of their main influencing factors, Hefang Street business block and Gongchen Bridge life block in Hangzhou are selected here as the study objects. Both non-participatory and participatory research methods are adopted to collect pedestrian information and observe pedestrians' ambiguous behavior, specific behavior, and stopping behavior. According to the study result, walking preference, walking time, environmental characteristics, and land-use mix (LUM) significantly impact pedestrian walking motivation. The type differences between Gongchen Bridge life block and Hefang Street business block leads to the difference in pedestrians' behaviors and their stopping time in business. Meanwhile, gender differences bring pedestrians' significant differences in walking motivation. Pedestrian walking preference and walking time are positively correlated with walking motivation in both streets. Environmental characteristics and LUM have also been proved to be important influencing factors of pedestrians' walking motivation. In this article, design and planning strategies are proposed for streets of different types in an attempt to provide reference for the revitalization and utilization of cultural heritage streets.

**Keywords:** historical and cultural streets; walking experience; walking behavior; public health

## 1. Introduction

The urban street is a necessary part of the city. Since ancient times, urban streets have reflected citizens' living standards, and related behavioral activities have been conducted by street pedestrians in the street space. This serves as a big promoter of public health. Today, as material civilization becomes increasingly mature, the streets morph into the dividing line between administrative areas, commercial areas, living areas and traffic areas, as well as an important carrier for functions of leisure, commerce, and entertainment. Therefore, the planning, construction and management of urban streets have become subjects worthy of key consideration. Studies on urban streets have been initiated at an early time, with a wide range of areas involved. As early as 1961, Jane Jacobs [1] in "The Death and Life of Great American Cities" conducted studies and made recommendations from the perspective of maintaining urban diversity and vitality. It's proposed that streets and sidewalks should be the main public areas in cities, and an in-depth analysis of the street's vitality and safety was conducted. Moreover, researchers extensively analyze the street space and its impact on pedestrian walking behavior and walking experience within the space from two dimensions: pedestrian's own conditions and environments. The impact of spatial environment on pedestrian satisfaction is explained via assessment and analysis [2,3]. Other studies have been carried out based on the relationship between people and streets.

Ways of creating spatial environments are provided via the study on pedestrian needs, the evaluation on the quality of pedestrian spaces, as well as the development and optimization of spatial design strategies [4–7].

In the study, Hangzhou of Zhejiang Province, a city with rich history and culture, is taken as the research background. Here, Hefang Street business block, and Gongchen Bridge life block are selected in Hangzhou for comparative analysis. These two streets are chosen for the case study because these two, in the same administrative district, exhibit different spatial and economic patterns of streets, thus the author can conduct an in-depth study on the pedestrian behavior, walking experience and their influencing factors in different blocks. An in-depth analysis is made on the street's history, passenger flow, spatial form, business distribution, and public facilities to lay the foundation for a study on analysis methods of these two blocks' differences. Literature from home and abroad is analyzed from four perspectives: pedestrian walking behavior, walking experience, pedestrian cities, and pedestrian streets, as well as the relationship between walking and public health, thus knowing the current situation and shortcomings of existing studies. Meanwhile, the pedestrian information is collected and analyzed through questionnaires. These questionnaires are distributed on weekdays, and followed up on weekends, with a view to obtaining as many samples as possible. Finally, 100 valid samples are collected from the streets. The basic information of pedestrians in these two streets is analyzed. In addition, SPSS data software is used to analyze the relationship between age and pedestrian ambiguous behavior, specific behavior and business stopping behavior. Besides, in the article, the impact of pedestrians' age, education level, walking preference, walking motivation, environmental characteristics and LUM on pedestrian walking time, is discussed, and the key factors impacting pedestrians' walking time are analyzed.

From the study perspective, the environmental assessment of the historic district is conducted by the observation of pedestrian walking behavior. In terms of study methodology, both participatory and non-participatory research and analyses are proposed to provide a basis for street researches of the same type. As for study application, life block and business block in historical streets are selected for comparison. The case selection containing cultural attribute, provides reference for the revitalization and utilization of cultural heritage streets from home and abroad.

## 2. Literature Review

### 2.1. Pedestrian Walking Behavior and Experience

While studying pedestrian walking behavior—a hot topic in recent years, some researchers have discussed it from different perspectives and multifaceted viewpoints based on the division of different pedestrian populations, largely in an attempt to figure out factors affecting walking, and study walking safety. For example, Ross [8], via the documented observations of relationship between children's walking behavior and the factors associated with it, discussed the impact of children's gender and walking time on their walking behavior. In the study of the built environment and walking behavior, Mirzaei [9] argued that most previous studies on walking behavior have focused on utilitarian or recreational walking behavior. Considering the differences in walking purposes, she explored the different effects of the built environment on walking behavior, thus confirming the necessity of study on walking motivation. Marisamynathan [10] discussed the influence of personal information, income, and road facilities on walking behavior of pedestrians crossing the road by studying the factors that affect walking behavior of pedestrians crossing the road, providing a method by which walking behavior of pedestrians crossing the road and its safety levels can be predicted. While conducting a comprehensive study of factors affecting walking behavior of pedestrians crossing the road, Aghabayk [11] discussed the differences between the signalized crossing and the unsignalized one, and examined overall the effects of gender, age, crossing awareness, technical equipment, and carried items on pedestrian crossing behaviors at the signalized crosswalk and the unsignalized one. In urban life, there are many road accidents involving pedestrian behavior that plays a central

part in these accidents. Jay Mathilde [12] studied the walking behavior of French and Japanese populations based on their differences, with a view to analyzing accidental risks caused by pedestrian behavior and providing selections for safe road crossing behavior. Mukherjee [13], from the survey data about video images from the signalized crossings, extracted individual pedestrian acts in violation of regulations and individual pedestrian road crossing behaviors, thereby predicting possible accidents.

The perspective of study on walking experience now is generally the combination of environmental psychology, walking experience and modern technology. Simulation and analysis and more are generally adopted as study methods [14,15]. The study content is divided into two parts: study and definition on the types of pedestrians [16], and study from perspectives including the intervention of the physical environment, the impact on pedestrians' psychological experience, and the humanitarian for special populations. For example, Bornioli, Anna [17], in the article, explored whether the physical environment affects the pedestrian walking experience and pedestrian psychology based on theories related to environmental psychology, concluding that the physical environment affects the pedestrian sensory experience- a key element of the walking experience. Then strategies can be established to create a good sensory experience through the construction of the physical environment. Stevenson and others [18] investigated the pedestrian walking experience via the interviews, as well as conceptualized and captured the pedestrian walking experience, in a bid to help pedestrians further deepen sense of leisure experience in a dynamic space, connect their physical environment, and bodily and mental environment to others, as well as strengthen their experience of the space through a wide range of connections. JIYOUNG [19] studied the negative pedestrian walking experience caused by subway stations by collecting materials, as well as analyzing and examining the function of subway stations. Here, the implication of walking as an experience was redefined, with the study purpose of turning a subway station into a positive space and creating a good pedestrian experience through music, sensors, and interesting facilities. Wong, Jeremy D [20] proposed that the preferred gait during walking can be adjusted by the nervous system to facilitate the reduction of the physical fatigue that pedestrians experience while walking, and the enhancement of the pedestrian's walking status from a neurological perspective. Cambra, Paulo [21] studied the interventions and effects of the built environment on adult walking behavior by modifying the physical environment, and also conducted a postmortem analysis. The study suggested that environmental interventions serve as an important factor influencing walking behavior, and possibly small-scale interventions in the walking environment can more effectively improve the walking experience. Tz-Yang Chao [22] improved the walking experience by adopting mixed reality technology. Pedestrians were guided to walk and interact with virtual characters within a prescribed range of mixed virtual reality technology, providing a new paradigm for the improvement of the walking experience. Ohjisuck [23] provided a realistic walking experience for the visually impaired by creating a new virtual walking experience environment, and setting up walking paths and braille devices, in the study on the walking experience of special populations from a sociological perspective. Jiyoung, Kwahk [24] developed pedestrian experience guidelines by analytical research and interviews with special populations with impairments in hearing, vision, speech, and physical mobility. These guidelines were used as a reference standard for the design and evaluation of pedestrian friendliness in pedestrian environments.

### 2.2. Cities and Pedestrian Streets

Based on urban pedestrian planning, the study is conducted from aspects of transportation, economy, and policy in the literature of this category. In these studies, the effective use of public transportation systems and information and communication technologies are advocated; innovations are made in implementation methods; and valuable proposals for urban economy, transportation, and environmental sustainability as well as the improvement of living spaces for urban residents, are offered based on the pedestrian city concept. For instance, Varma [25] suggested that the effective use of public transport

systems and pedestrian cities should be becoming a priority for urban development. Varma also explored the impact of new trends in urban mobility and information and communication technologies on the cities' future development. With Seoul as an example, Young, Kim Sun, and others [26] analyzed the gait features of pedestrians by photographing walking streets in order to find the relationship between the rational selection of pedestrians and the extracted walking environment. Rebecchi [27] proposed a framework for assessing the walkability of cities to study the strengths and weaknesses of the urban environments and to improve healthy living spaces. Yassin [28] proposed an innovative practice specially used for pedestrians, in a bid to re-attract people to the downtown and pedestrian environment. Sustainable urban development is realized by the restoration of urban walkability. Fallahranjbar [29] proposed that putting people first should be a key development principle for all cities that aim to improve urban quality; and a healthy environment should be fundamentally related to a pedestrian community. Fancello [30] explored how citizens' preferences and values vary spatially, as well as designed walking policies that improve citizens' quality, providing new recommendations for mapping out walkability-oriented urban policies. Hui He [31] pointed out that population density was negatively correlated with sport frequency and total sport time, and proposed intervention strategies for an aging-appropriate urban environment. The research focus was the relationship between vehicle conditions, visual signals such as monitors and environmental perception signals, and walking decisions. Fanny Malin [32] evaluated the short- and long-term impact of speed display signs in pedestrian street on the speed of moving vehicles in a low-speed urban environment. If the speed displays are installed at pedestrian crossings, the speed of moving vehicles will be reduced, and pedestrian safety will be guaranteed. With historical street of Shapowei, Xiamen, as an example, Lemin Zhang [33] constructed a street vitality evaluation system based on spatio-temporal data of pedestrians, as well as systematically examined the impact of the built environment on street vitality in historical streets using multi-modal analysis techniques.

### 2.3. Impact of Walking on Public Health

Considering the inevitable link between walking and public health, there are increasing studies on how urban transportation planning meets the need of public health. Most literatures focus on the benefits of a walking lifestyle for public health, including physical and mental benefits. On this basis, the discussion is made based on specific influencing factors in these literatures. Also, some researchers have studied the adverse effects of walking on public health by attributing them to the external environments on basis of a comprehensive consideration. The positive impact of walking on health has been studied mainly from internal factors. For instance, D. Merom [34] argued that less dependence on cars facilitates public health, and that active behavioral activities provide a solution to sedentariness and lack of physical activities. James F Sallis [35] concluded from his analysis that most environmental attributes are positively correlated with physical activities. The study results are similar in each and every city. James pointed out, the design of urban environments contributes a lot to physical activities, and the global health burden brought by physical inactivity can be reduced through the participation of all departments. Other researchers discussed on basis of specific walking-based influencing factors of the health benefits. Mohammad Javad Koohsari [36] noted that, public open space brings many health benefits physically and mentally. In the study, multilevel logistic regression models are adopted to examine the correlation between measured values of public open space and walking and depression, and the importance of the potential impact of these assessment criteria on health is emphasized. By conducting experiments, Ming-YiHsu [37] proved that brisk walking, as an effective physical activity promoting mental health in adolescents, can reduce depression and anxiety as well as improve self-assessment. However, the walking lifestyle is a double-edged sword for health. The negative impacts are mainly studied from external factors, such as air quality. Giorgos Giallouros [38] argued that, commuting by walking, compared with vehicles, may increase the intake of fine particles for pedestrians,

and cause negative health effects. The arguments were started from an integrated relative risk perspective. Caihua Zhu [39] et al. thought, PM2.5 in the air impact physical health when pedestrians are fully exposed to the outside environment. They studied the specific impacts as a way to evaluate the risk features of walking and provide recommendations for the improvement of residents' health and decision making about walking trips.

It's found from the analysis of the literature that. Two main features of the existing studies are as follows:

The study is conducted on external factors of pedestrians. Pedestrians are impacted by many factors amid walking. Especially, the physical environment can directly impact pedestrians' physical and psychological experiences. Thus it can be said that pedestrian walking experience is related to the physical environments (roads, signals, facilities, etc.). It is confirmed that, amid the street construction, it is necessary to provide an excellent physical and sensory experience for pedestrians via the construction of the physical environment on the one hand. The impact of the physical environment on different groups, as well as their applicability in the environment, need to be taken into account, thereby meeting the walking experience of people of different types on the other hand. On this basis, the factors influencing the pedestrian walking experience are further investigated via the interventions and technological means that change the physical environment.

The study on the pedestrian ontology. On the one hand, it's confirmed physiologically and psychologically that a walking lifestyle can improve the emotional state, thus promoting public health. Amid construction, physical and mental impact on the public should be considered, and the negative impact from external environment should be minimized. An all-round street space environment, which facilitates daily traveling, and promotes public health, should be constructed. On the other hand, the study is conducted on the division of pedestrian types, and pedestrian behavior differences; the construction of a sustainable city is explored from the urban walkability; policies based on walking city are advocated; importance is attached to the planning of walking environment; citizens' walking behavior is guided. However, in most existing studies, non-participatory pedestrian behavior is referred, and attention is more likely to be paid to pedestrians' behavior and experience in a single physical environment. While the subjective intention of pedestrians is neglected. The policies about street design and pedestrian urban planning are proposed without support of enough studies on pedestrian ontology. Additionally, there is still much room for improvement in comparative studies on walking behaviors in different places. Therefore, in the study, both non-participatory and participatory approaches are adopted, with emphasis on the subjective evaluation of pedestrians. Pedestrian behavior states on streets of different types are observed and recorded, and a multi-dimensional study of pedestrian walking behavior is conducted, so as to put forward constructive suggestions in a targeted manner.

## 3. Methods and Measures

Literature from home and abroad is analyzed to learn the current situation and shortcomings of existing studies. In the study, two blocks-Hefang Street business block and Gongchen Bridge life block are selected. The mainly investigated roads in Hefang Street include the main road of Hefang Street, Qinghefang Community, Gaoyin Street, Huimin Road, Dingan Road, and Huaguang Road. The mainly investigated roads in Gongchen Bridge involve Xiaohelu Community, Lishui Road, and Scenic Street Area. Here major factors impacting pedestrian behaviors are explored in the block.

The investigation is divided into two parts: field investigation and user investigation. The former consists of two parts: site investigation and behavior observation. Interview, photo-taking and other forms are adopted to obtain the distribution information of street businesses, including the distribution locations of businesses of different types and the proportion of the business of a certain type in the business as a whole. In terms of building scale and street paving and decoration, the building height, road width and paving material are investigated and analyzed. While behavior observation activity is performed, pedestrian

behavior is observed and recorded in tracking investigation, that is, pedestrians are tracked in real time in the street and their status and staying time in shops are recorded. Different business types include essential necessities, commercial consumption facilities, leisure and entertainment facilities and public service facilities. In the user survey, pedestrians, staff of surrounding shops, recreational residents in the community and other street users are invited to fill out questionnaires for information collection. While some of pedestrians with poor educational background or the aged ones can't read the questionnaires, so they are inquired orally and recorded. The questionnaire contents include pedestrian behaviors, socio-demographic characteristics, total walking time, walking preference, walking motivation and environmental characteristics. Information including the participants' demographic data, their environment satisfaction and daily travelling behavior preferences are obtained in the questionnaire of the study (Figure 1).

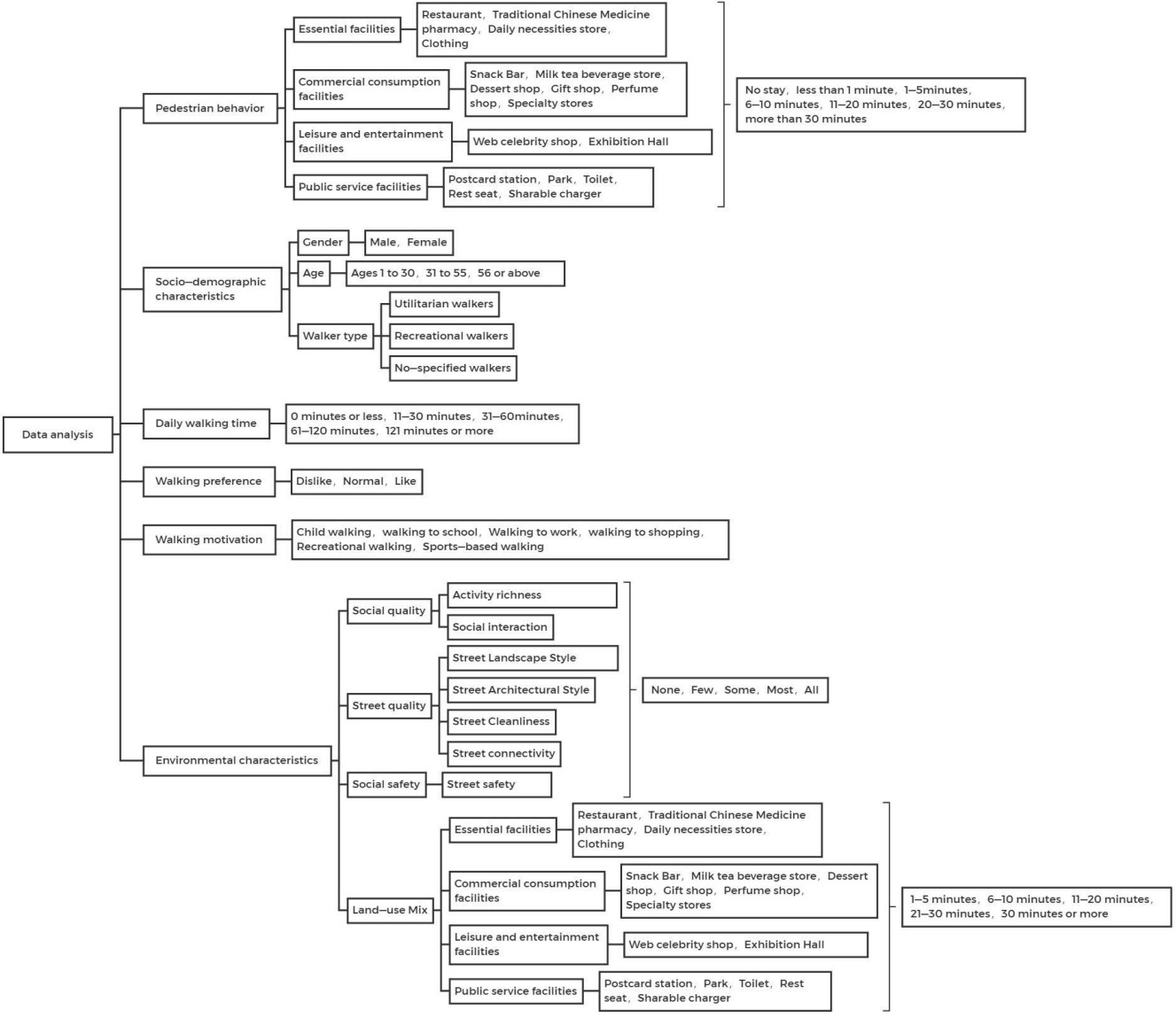

**Figure 1.** Data analysis framework.

"Socio-demographic characteristics" include gender, age, education, profession, and income. There are two categories of gender: male and female. The ages are divided into three categories: 1–30, 31–55, 55 or above. Street walkers are divided into utilitarian walkers, recreational walkers, and no-specified walkers. Utilitarian walkers mainly walk for commuting between home and school or workplace. Recreational walkers mainly walk

and jog. The no-specified walkers mainly walk for daily affairs, such as going shopping, going to hospital, visiting friends, and doing some commute-related things.

The specific businesses where pedestrians stay are: (1) essential necessities such as restaurants, traditional Chinese medicine pharmacies, daily necessities stores, and clothing stores; (2) commercial consumption facilities such as snack bars, milk tea beverage stores, dessert shops, gift shops, perfume shops, and specialty shops; (3) leisure and entertainment facilities such as web celebrity shops, and exhibition halls; (4) public service facilities: postcard stations, parks, toilets, rest seats, and sharable chargers.

"Total walking time" is assessed by individual items, with a 5-point system (1 = 10 min or below; 2 = 11–30 min; 3 = 31–60 min; 4 = 61–120 min; 5 = 121 min or above) adopted to measure respondents' "daily walking time".

"Walking preference" is assessed by an individual item, with a 3-point system (1 = "dislike"; 2 = "normal"; 3 = "Like") adopted to answer the question "How much do you enjoy walking on a daily basis?".

"Walking motivation" is assessed by six items, with a 5-point system adopted to measure (1 = none; 2 = few; 3 = some; 4 = most; 5 = all), and six categories include "walking with children", "walking to study", "walking to work", "walking to shop", "taking a walk to play", and "walking exercise".

"Environmental characteristics" refer to four dimensions, of which social quality, street quality and safety issues, are scored based on a 5-point system (1 = none; 2 = minority; 3 = some; 4 = majority; 5 = all). For the LUM, the 5-point system (1 = 1–5 min; 2 = 6–10 min; 3 = 11–20 min; 4 = 21–30 min; 5 = over 30 min) is adopted to measure the walking time of respondents using facilities of 4 types. Here, social quality refers to two dimensions: namely, "activity richness" and "social interaction degree"; street quality includes four dimensions: namely, street landscape style, street architectural style, street cleanliness and street connectivity; safety issue is street safety; LUM involves four dimensions: i.e., essential facilities, commercial consumption facilities, leisure and entertainment facilities, and public service facilities.

The LUM consists of 17 items used to assess land-use mix, "LUM (Land-use Mix)"and a 5-point system (1 = 1–5 min; 2 = 6–10 min; 3 = 11–20 min; 4 = 21–30 min; 5 = over 30 min) is adopted to measure respondents' walking time in 17 amenities, with lower scores indicating higher levels of land-use mix. The 19 items of LUM are grouped into four dimensions: (1) essential facilities; (2) commercial consumption facilities; (3) leisure and entertainment facilities; and (4) public service facilities.

## 4. Results

### 4.1. Historical Evolution

4.1.1. The Historical Evolution of Hefang Street and Gongchen Bridge Life Block

In this study, the Hefang Street business block and the Gongchen Bridge life block, both of which are located in the main city zone of Hangzhou, are the study subjects. These two differ in street forms and economic patterns.

As shown in Figure 2, Hefang Street located at the foot of Wushan, is part of Qinghefang and belongs to the old city of Hangzhou. Located in the northern part of Hangzhou city, the whole Gongchen Bridge is built above the ancient Beijing-Hangzhou Grand Canal, mainly including two parts: Gongchen Bridge block and the historical and cultural block in the west of the bridge.

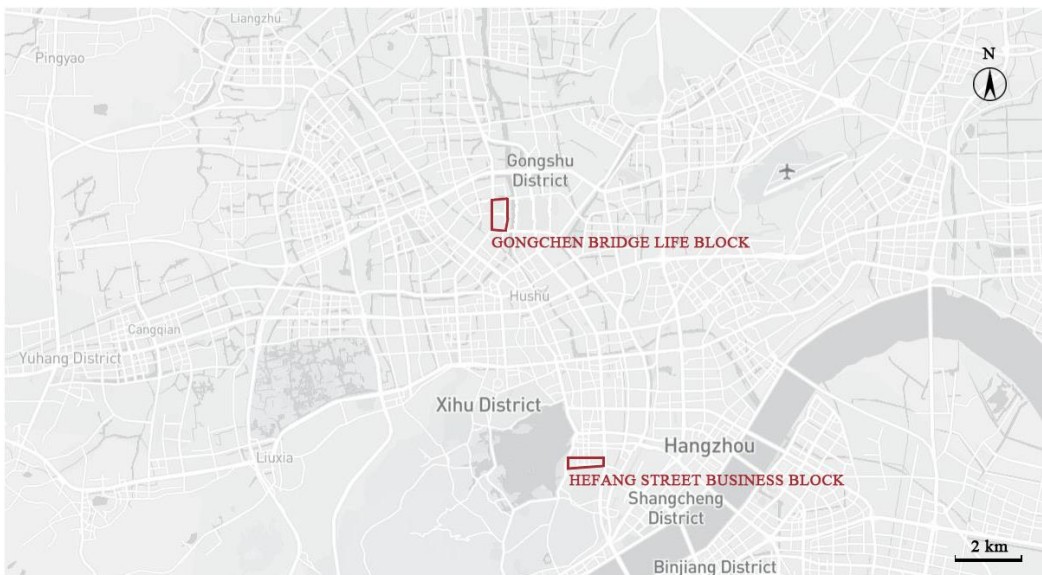

**Figure 2.** Location map of Hefang Street business block and Gongchen Bridge life block (according to Baidu map).

Once upon a time, Hefang Street was the "root of the imperial city" of Hangzhou, the capital of ancient times, and also the cultural center and economic and trade center of the Southern Song Dynasty. As shown in Figure 3, as the only old street in Hangzhou that maintains the historical appearance of the old city, Hefang Street, where time-honored brands (stores) stood in great numbers, integrated the most representative historical culture, commercial culture, marketplace culture and architectural culture in Hangzhou, so that it was saved from the fate of total demolition during the transformation of the old city of Hangzhou.

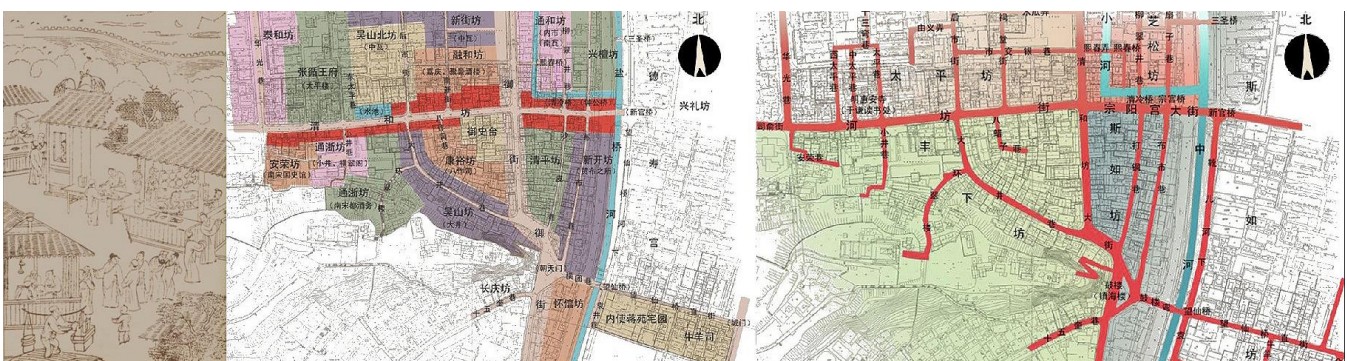

**Figure 3.** paintings in the past (**left**), Street pattern map of southern Song Dynasty (**middle**) and Street pattern map of Ming and Qing Dynasties (**right**). (online website https://jz.docin.com/p-2811 341437.html accessed on 22 November 2021).

In 1999, the Hangzhou Municipal Government decided to redevelop the 13.6-hectare block adjacent to the Wushan (Town God's Temple) square into an antique pedestrian street of commerce and tourism. The renovated Hefang Street was opened in October 2002, as shown in Figure 4. Nowadays, Hefang Street reveals the style of the late Qing Dynasty and early Ming Dynasty. The cultural value is highlighted here to create a marketplace culture of commerce, pharmaceutical industry, and architecture. Its historical authenticity, cultural continuity and the overall appearance of the landscape are maintained. On both sides of Hefang Street, there are mainly local products, antiques, paintings and tourist souvenirs, and a craft pavilion is set up in the center of the street to display folk handicraft performances.

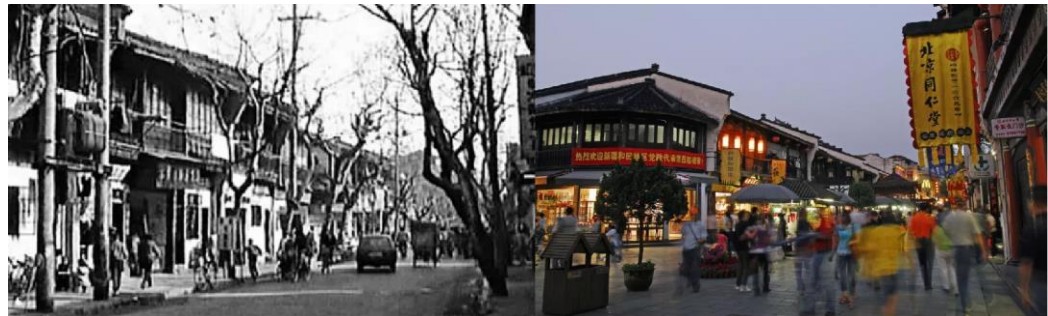

**Figure 4.** Photos of Hefang Street in 1966 (**left**) and the current (**right**). (online website http://www.360doc.com/content/20/0812/10/17132703_929812656.shtml accessed on 25 November 2021).

Now, there are many problems in Hefang Street. First of all, the public design at the monotonous node of the entrance space of Hefang Street is relatively simple. Most of them are seats combined with tree pools and simple humanoid sculptures. They are not attractive to walkers. The whole street is similar in format and has a high repetition rate. The feeling is bland and cannot produce great fluctuations in the sensory psychology of walkers. The viewing speed of respondents has not changed significantly. Secondly, in the design process of Hefang Street, there is not enough leisure space at the door of the shop which is easy to arouse pedestrians' interest, and the influence of natural factors such as season and climate is not considered, which is easy to cause local space congestion due to natural factors such as sunshine and rain, which affects pedestrians' walking experience. Finally, due to the high degree of commercialization of blocks in the old city bring vitality, the excessive commercialization and destruction of the traditional charm of space in Hefang Street reflects the traditional culture. Except for the architectural appearance, most of them show the traditional life scene by sculpture. Such facilities cannot interact with pedestrians, cannot render the traditional cultural atmosphere of the street, cannot leave a deep impression on pedestrians, and lack of interactive interest.

The relatively desolate original site of Gongchen bridge featured by a dotted scattering of the street layout was later evolved into the terminal dock of the Grand Canal in ancient times(Figure 5). The street was developed linearly along the river and extended radially to the interior. The Qing government, after the Sino-Japanese War, signed the Treaty of Shimonoseki with the Japanese imperialists, and Gongchen bridge was set up as a Japanese concession. Later, as the renovation project of the old city of Gongchen bridge area advanced, the faceted blocks and grouped factories took shape. In modern times, Gongchen bridge becomes a cultural preservation unit of Zhejiang Province, and also a part of the downtown. Unlike Hefang Streat with a mature business block mode, Gongchen Bridge, as a living pedestrian street, mainly serves pedestrians and surrounding residents.

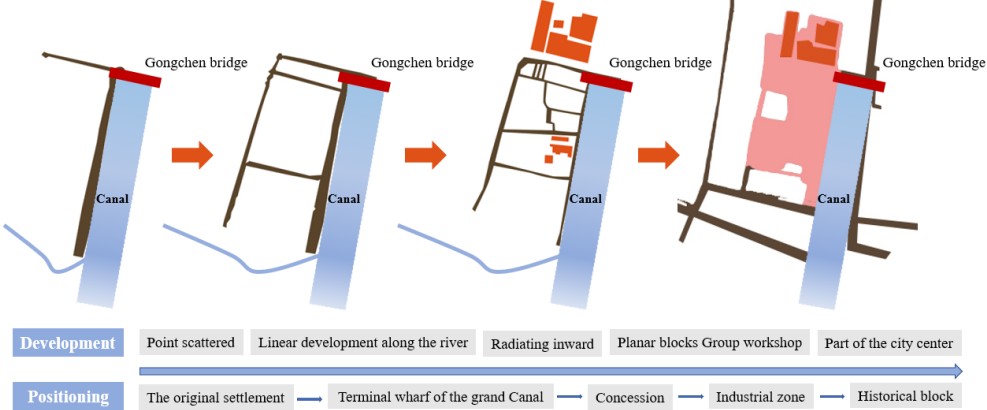

**Figure 5.** Morphological evolution of Gongchen Bridge Block. (Self-painted according to the historical evolution of Gongchen Bridge).

At present, the main problems of Gongchen Bridge block are that some houses were built earlier, and the street landscape environment quality in some roadways was not good. Because of the lack of protection awareness of historical and cultural blocks, the color volume of newly built buildings could not be well integrated with the style of historical and cultural blocks, which seemed to be somewhat out of place. Secondly, due to the high population density and mostly elderly people, the problem of aging in streets is prominent, and the blocks lack vitality and provide leisure and entertainment facilities are also in short supply.

### 4.1.2. Block Spatial Scale

In the form of street space, two streets have different forms of site distribution (Figure 6).The ratio of distance (D) and height (H) of building along Hefang Street-D/H = 13/6 = 2.167 > 2. The less closed and more open space as a whole is spacious and low, with an overall trend of horizontal extension generating a sense of openness and flow, but less affinity. Therefore, store seats in Hefang Street have been set up in the middle section of the road, increasing pedestrians' sense of safety for this place and creating spacious human touch in the block.

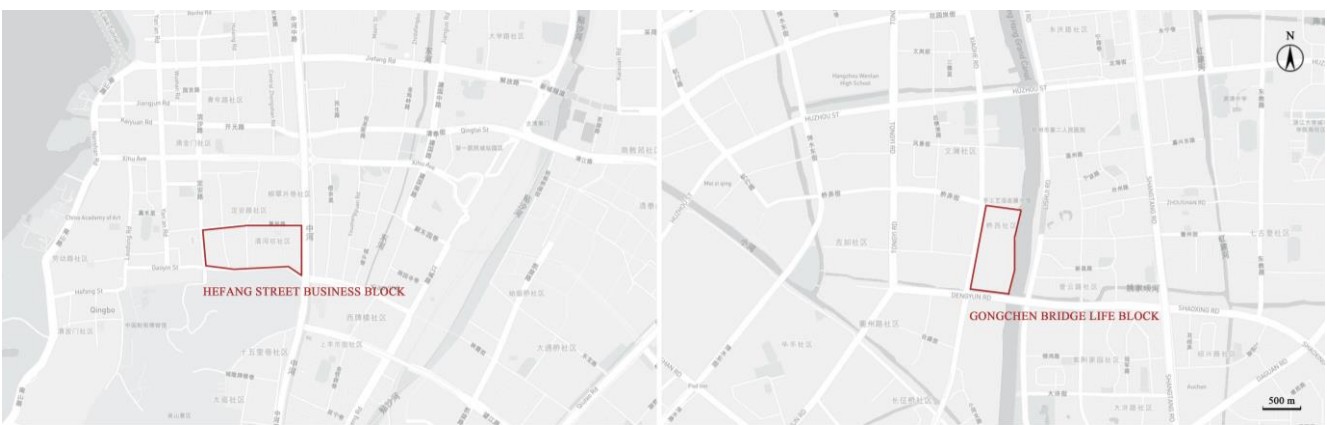

**Figure 6.** Maps of Hefang Street business block and Gongchen Bridge life block (according to Baidu map).

The streets in historical Gongchen Bridge block are mainly divided into two types-Qiaoxi Straight Street—a commercial block and Jixiang Temple Alley-a living alleyway. The ratio of distance (D) and height (H) of buildings along the commercial block—Qiaoxi Straight Street is D/H = 6/9 = 0.667 < 1, and the ratio of distance (D) and height (H) of buildings along the living alleyway Jixiang Temple Alley is D/H = 2.5/8 = 0.3125 < 1. It can be seen that the overall space of Gongchen Bridge block brings a strong sense of closure, and the long and narrow street creates a sense of profundity in the space.

### 4.1.3. Block Business Distribution

In Gongchen Bridge life block, there are totally 9 essential facilities accounting for 11%, 14 commercial consumption facilities taking up 20%, 10 leisure and entertainment facilities accounting for 13%, and 41 public service facilities taking up 55%. In Hefang Street business block, there are totally 63 essential facilities accounting for 20.3%, 109 commercial consumption facilities taking up 35%, 16 leisure and entertainment facilities accounting for 5.1%, and 123 public service facilities taking up 39.5%. In terms of essential facilities, restaurants play the biggest part in both Gongchen Bridge and Hefang Street, while there is only one daily necessities store in Gongchen Bridge. There is the same number of Traditional Chinese Medicine pharmacies in both of them. In terms of commercial consumption facilities, gift shops (5 in Gongchen Bridge and 65 in Hefang Street) play the biggest part in both of them. There is only one snack bar, one perfume shop and one specialty store in Gongchen Bridge, while there are 13 snack bars, 19 specialty stores, and only 1 dessert shop in Hefang Street. In terms of leisure and entertainment facilities, there is almost the same number of web celebrity shops in these two. There are 11 exhibition halls in Hefang Street and there is 6 in Gongchen

Bridge. In terms of public service facilities, there are 15 rest seats, and 22 sharable chargers in Gongchen Bridge, and 49 rest seats and 70 sharable chargers in Hefang Street. The number of postcard stations, parks and toilets is the same in these two.

According to the statistics of the number and proportion of businesses in the two blocks (Table 1), Hefang Street business block, relatively, boasts rich businesses, and the proportion of facilities of different types is balanced. Only leisure and entertainment facilities play a very small part there. Essential necessities and commercial consumption facilities in Hefang Street business block, outnumber that in Gongchen Bridge life block. Public service facilities have the highest proportion in Gongchen Bridge life block, while essential facilities account for the least there, so that the daily needs of the surrounding residents can't be met. In terms of the proportion of leisure and entertainment facilities and public service facilities, Gongchen Bridge outperforms Hefang Street.

**Table 1.** Number formats of Hefang Street business block and Gongchen Bridge life block.

| Type | Formats | Gongchen Bridge Life Block (n) | Gongchen Bridge Life Block (%) | Hefang Street Business Block (n) | Hefang Street Business Block (%) |
|---|---|---|---|---|---|
| Essential facilities | Restaurant | 4 | 5% | 50 | 16% |
| | Traditional Chinese Medicine pharmacy | 3 | 4% | 3 | 1% |
| | Daily necessities store | 1 | 1% | 8 | 2.5% |
| | Clothing | 1 | 1% | 2 | 0.6% |
| Total | | 9 | 11% | 63 | 20.3% |
| Commercial consumption facilities | Snack Bar | 1 | 1% | 13 | 4.2% |
| | Milk tea beverage store | 3 | 4% | 9 | 2.9% |
| | Dessert shop | 3 | 4% | 1 | 1% |
| | Gift shop | 5 | 10% | 65 | 21% |
| | Perfume shop | 1 | 1% | 2 | 0.6% |
| | Specialty stores | 1 | 1% | 19 | 6% |
| Total | | 14 | 20% | 109 | 35% |
| Leisure and entertainment facilities | Web celebrity shop | 4 | 5% | 5 | 1.6% |
| | Exhibition Hall | 6 | 8% | 11 | 3.5% |
| Total | | 10 | 13% | 16 | 5.1% |
| Public service facilities | Post card station | 1 | 1% | 1 | 1% |
| | Park | 1 | 1% | 1 | 1% |
| | Toilet | 2 | 3% | 2 | 0.6% |
| | Rest seat | 15 | 20% | 49 | 15.8% |
| | Sharable charger | 22 | 30% | 70 | 23% |
| Total | | 41 | 55% | 123 | 39.5% |

*4.2. Basic Pedestrian Information*

Descriptive statistics of pedestrians in both blocks are shown in Table 2. The number of male respondents is almost equal to that of female respondents in Hefang Street, with females accounting for 52% and males taking up 48%. Pedestrians aged 1–30, the main part of pedestrians, account for 54.7% of all, followed by those aged 31–55 accounting for 31.3%; those aged 56 and above account for only 14%. Pedestrians in Hefang Street are mainly recreational walkers, accounting for 43.3% of the total, while utilitarian walkers play a small part, taking up only 16% of the total. Among respondents in Gongchen Bridge, females account for 43.3% and males 56.7%. Those aged 31–55, the main part of respondents, account for 56.7% of all, followed by those aged 1–30 taking up 30%; those aged 56 and above, as the small part of the whole, only account for 13.3%. Pedestrians in Gongchen Bridge are mainly recreational walkers accounting for 55.3% of the total, followed by utilitarian walkers taking up 25.3%. No-specified walkers account for only 19.3% of the total.

**Table 2.** Pedestrian information in Hefang Street business block and Gongchen Bridge life block.

| Pedestrian Information | Describe | GCQ | | HFJ | |
|---|---|---|---|---|---|
| | | n | % | n | % |
| All | All | 150 | 100% | 150 | 100% |
| Gender | Male | 85 | 56.7% | 72 | 48% |
| | Female | 65 | 43.3% | 78 | 52% |
| Age | 1–30 | 45 | 30% | 82 | 54.7% |
| | 31–55 | 85 | 56.7% | 47 | 31.3% |
| | 56 or above | 20 | 13.3% | 21 | 14.0% |
| Pedestrian type | Utilitarian walkers | 38 | 25.3% | 24 | 16% |
| | Recreational walkers | 83 | 55.3% | 65 | 43.3% |
| | No-specified walkers | 29 | 19.3% | 61 | 40.7% |

*4.3. Pedestrian Behavior Analysis*

1. The data about different genders and business staying time show that (Table 3):

**Table 3.** Mean and standard deviation of dwell time for pedestrian street furniture of different genders.

| Gender | The Overall | | | | Male | | | | Female | | | |
|---|---|---|---|---|---|---|---|---|---|---|---|---|
| | HFJ | | GCQ | | HFJ | | GCQ | | HFJ | | GCQ | |
| | N = 150 | | N = 150 | | N = 72 | | N = 85 | | N = 78 | | N = 65 | |
| | Mean | SD | Mean | SD | Mean | SD | Mean | SD | Mean | SD | Mean | SD |
| Pedestrian behavior (stay) | | | | | | | | | | | | |
| Essential facilities | | | | | | | | | | | | |
| Restaurant | 0.880 | 1.634 | 2.200 | 2.288 | 1.014 | 1.756 | 2.318 | 2.341 | 0.756 | 1.513 | 2.046 | 2.225 |
| Traditional Chinese Medicine pharmacy | 0.860 | 1.810 | 1.693 | 2.397 | 0.861 | 1.739 | 1.788 | 2.508 | 0.859 | 1.884 | 1.569 | 2.257 |
| Daily necessities store | 0.520 | 0.981 | 0.947 | 1.310 | 0.597 | 1.057 | 0.765 | 1.151 | 0.449 | 0.907 | 1.185 | 1.467 |
| Clothing | 0.927 | 1.457 | 1.540 | 1.709 | 1.083 | 1.545 | 1.141 | 1.465 | 0.782 | 1.364 | 2.062 | 1.870 |
| Commercial consumption facilities | | | | | | | | | | | | |
| Snack Bar | 1.267 | 1.294 | 1.453 | 1.256 | 1.403 | 1.411 | 1.447 | 1.305 | 1.141 | 1.170 | 1.462 | 1.200 |
| Milk tea beverage store | 0.627 | 1.127 | 1.093 | 1.318 | 0.667 | 1.163 | 1.071 | 1.289 | 0.590 | 1.098 | 1.123 | 1.364 |
| Dessert shop | 0.887 | 1.156 | 0.973 | 1.080 | 0.806 | 1.182 | 1.106 | 1.124 | 0.962 | 1.133 | 0.800 | 1.003 |
| Gift shop | 1.507 | 1.646 | 1.767 | 1.826 | 1.583 | 1.642 | 1.729 | 1.880 | 1.436 | 1.656 | 1.815 | 1.767 |
| Perfume shop | 0.507 | 0.910 | 0.747 | 1.112 | 0.583 | 1.071 | 0.494 | 0.840 | 0.436 | 0.731 | 1.077 | 1.327 |
| Specialty stores | 0.967 | 1.234 | 0.887 | 1.308 | 1.056 | 1.299 | 0.882 | 1.331 | 0.885 | 1.173 | 0.892 | 1.288 |
| Leisure and entertainment facilities | | | | | | | | | | | | |
| Web celebrity shop | 0.993 | 1.445 | 1.593 | 1.655 | 1.125 | 1.501 | 1.612 | 1.655 | 0.872 | 1.390 | 1.569 | 1.667 |
| Exhibition Hall | 1.787 | 2.298 | 2.827 | 2.716 | 1.819 | 2.222 | 3.106 | 2.699 | 1.756 | 2.381 | 2.462 | 2.716 |
| Public service facilities | | | | | | | | | | | | |
| Post card station | 0.867 | 1.544 | 1.653 | 1.791 | 1.000 | 1.653 | 1.835 | 1.883 | 0.744 | 1.436 | 1.415 | 1.648 |
| Park | 0.993 | 1.679 | 1.887 | 1.859 | 1.042 | 1.736 | 1.941 | 1.911 | 0.949 | 1.635 | 1.815 | 1.802 |
| Toilet | 0.480 | 0.857 | 0.880 | 0.989 | 0.556 | 0.933 | 0.706 | 0.828 | 0.410 | 0.780 | 1.108 | 1.134 |
| Rest seat | 1.653 | 1.843 | 2.207 | 2.162 | 1.653 | 1.602 | 2.271 | 2.055 | 1.654 | 2.050 | 2.123 | 2.308 |
| Sharable charger | 0.253 | 0.657 | 0.380 | 0.720 | 0.333 | 0.751 | 0.400 | 0.727 | 0.179 | 0.552 | 0.354 | 0.717 |

SD—standard deviation. Pedestrians (staying): "0" means "no staying"; "1" means "within 1 min"; "2" represents "1–5 min"; "3" means "6–10 min"; "4" represents "11–20 min"; "5" means "20–30 min"; "6" means "over 30 min".

Generally, the staying time in all business types is for pedestrians in Hefeng Street is shorter than that in Gongchen Bridge. Pedestrians in Hefang Street have the longest stay in commercial consumption facilities (5.762), while those in Gongchen Bridge have the longest stay in public service facilities (7.007). In terms of pedestrian's staying time in essential facilities (Hefang Street = 3.187, Gongchen Bridge = 6.380), the difference in both

blocks is the most obvious. Pedestrians in both blocks have the shortest staying time in recreational facilities (Hefang Street = 2.780, Gongchen Bridge = 4.420).

Male pedestrians in Hefang Street spend the longest time in commercial consumption facilities (6.098), while the male pedestrians in Gongchen Bridge spend the longest time in public service facilities (7.153). In addition, there is an obvious difference between male pedestrians' staying time in public service facilities (Hefang Street = 4.584, Gongchen Bridge = 7.153) in the two. For example, the park staying time in Hefang Street (1.042) is less than that in Gongchen Bridge (1.941).

The female pedestrians in both blocks spend the longest time in commercial consumption facilities (Hefang Street = 5.450, Gongchen Bridge = 7.169), while there is the most obvious difference between female pedestrians in both blocks in terms of the time spent on essential necessities (Hefang Street = 2.846, Gongchen Bridge = 6.862). For example, pedestrians' staying time in restaurants in Hefang Street (0.756) is shorter than that in Gongchen Bridge (2.046).

2.  The data of age and business staying time (Table 4):

**Table 4.** Mean and standard deviation of dwell time for pedestrian street facilities of different ages.

| Age | 1–30 | | | | 31–55 | | | | 56 or Above | | | |
|---|---|---|---|---|---|---|---|---|---|---|---|---|
| | Hefang Street | | Gongchenqiao Street | | Hefang Street | | Gongchenqiao Street | | Hefang Street | | Gongchenqiao Street | |
| | N = 82 | | N = 45 | | N = 47 | | N = 85 | | N = 21 | | N = 20 | |
| | Mean | SD | Mean | SD | Mean | SD | Mean | SD | Mean | SD | Mean | SD |
| Pedestrian behavior (stay) | | | | | | | | | | | | |
| Essential facilities | | | | | | | | | | | | |
| Restaurant | 0.610 | 1.368 | 2.022 | 2.426 | 1.000 | 1.694 | 2.235 | 2.213 | 1.667 | 2.176 | 2.450 | 2.373 |
| Traditional Chinese Medicine pharmacy | 0.476 | 1.269 | 1.311 | 2.087 | 0.809 | 1.715 | 1.576 | 2.362 | 2.476 | 2.786 | 3.050 | 2.819 |
| Daily necessities store | 0.366 | 0.854 | 0.689 | 1.145 | 0.809 | 1.191 | 1.012 | 1.314 | 0.476 | 0.814 | 1.250 | 1.585 |
| Clothing | 0.793 | 1.438 | 1.667 | 1.895 | 1.085 | 1.501 | 1.541 | 1.666 | 1.095 | 1.446 | 1.250 | 1.482 |
| Commercial consumption facilities | | | | | | | | | | | | |
| Snack Bar | 1.378 | 1.330 | 1.644 | 1.317 | 1.213 | 1.250 | 1.365 | 1.184 | 0.952 | 1.244 | 1.400 | 1.429 |
| Milk tea beverage store | 0.476 | 0.933 | 1.089 | 1.240 | 0.766 | 1.088 | 1.000 | 1.185 | 0.905 | 1.729 | 1.500 | 1.906 |
| Dessert shop | 0.976 | 1.165 | 1.156 | 1.127 | 0.766 | 1.237 | 0.929 | 1.067 | 0.810 | 0.928 | 0.750 | 1.020 |
| Gift shop | 1.220 | 1.457 | 1.933 | 1.900 | 1.745 | 1.811 | 1.741 | 1.774 | 2.095 | 1.786 | 1.500 | 1.933 |
| Perfume shop | 0.585 | 0.942 | 1.111 | 1.210 | 0.511 | 0.997 | 0.635 | 1.056 | 0.190 | 0.402 | 0.400 | 0.940 |
| Specialty stores | 0.854 | 1.101 | 0.822 | 1.336 | 1.106 | 1.463 | 0.859 | 1.329 | 1.095 | 1.179 | 1.150 | 1.182 |
| Leisure and entertainment facilities | | | | | | | | | | | | |
| Web celebrity shop | 1.061 | 1.651 | 2.156 | 1.953 | 0.830 | 1.129 | 1.400 | 1.474 | 1.095 | 1.221 | 1.150 | 1.387 |
| Exhibition Hall | 1.500 | 2.218 | 3.000 | 2.820 | 1.617 | 2.202 | 2.800 | 2.689 | 3.286 | 2.348 | 2.550 | 2.704 |
| Public service facilities | | | | | | | | | | | | |
| Post card station | 0.927 | 1.661 | 2.289 | 2.030 | 0.681 | 1.337 | 1.376 | 1.647 | 1.048 | 1.532 | 1.400 | 1.501 |
| Park | 0.744 | 1.464 | 1.489 | 1.727 | 1.196 | 1.928 | 2.035 | 1.930 | 1.524 | 1.778 | 2.150 | 1.785 |
| Toilet | 0.427 | 0.847 | 0.822 | 0.936 | 0.489 | 0.804 | 0.906 | 0.971 | 0.667 | 1.017 | 0.900 | 1.210 |
| Rest seat | 1.390 | 1.762 | 2.289 | 2.052 | 1.809 | 1.789 | 1.953 | 2.098 | 2.333 | 2.129 | 3.100 | 2.511 |
| Sharable charger | 0.317 | 0.752 | 0.489 | 0.589 | 0.106 | 0.312 | 0.294 | 0.651 | 0.333 | 0.796 | 0.500 | 1.147 |

SD—standard deviation. Pedestrians (staying): "0" means "no staying"; "1" represents "within 1 min"; "2" means "1–5 min"; "3" represents "6–10 min"; "4" means "11–20 min"; "5" represents "20–30 min"; "6" means "over 30 min".

Pedestrians aged 1–30 in both blocks spend the longest time in commercial consumption facilities (Hefang Street = 5.489, Gongchen Bridge = 7.755). Pedestrians aged 1–30 in Hefang Street spend the shortest time in essential facilities (2.245), while pedestrians aged 1–30 in Gongchen Bridge spend the shortest time in the leisure and entertainment facilities (5.156). In addition, there is the most obvious difference in the staying time of pedestrians aged 1–30 in public service facilities (Hengfang Street = 3.805, Gongchen Bridge = 7.738) be-

tween these two. For example, pedestrians in Gongchen Bridge stay longer in the postcard station (2.289) than that in Hefang Street (0.927).

Pedestrians aged 31–55 in both blocks have the shortest staying time in leisure and entertainment facilities (Hefang Street = 2.447, Gongchen Bridge = 4.200). Pedestrians aged 31–55 in Hefang Street spend the longest time in commercial consumption facilities (6.107), while pedestrians aged 31–55 in Gongchen Bridge spend the longest time in public service facilities (6.564). Moreover, the most obvious difference can be seen in the staying time of essential facilities for pedestrians aged 31–55 in these two blocks (Hefang Street = 3.703, Gongchen Bridge = 6.364). For example, pedestrians stay longer in restaurants in Gongchen Bridge (2.235) than those in Hefang Street (1.000).

Pedestrians aged 56 and above in both blocks have the shortest stay time in leisure and entertainment facilities (Hefang Street = 4.381, Gongchen Bridge = 3.700). Pedestrians aged 56 and above in Hefang Street spend the longest time in commercial consumption facilities (6.047), while pedestrians aged 56 and above in Gongchen Bridge spend the longest time in public service facilities (8.050). Besides, there is the most obvious difference in staying time in essential facilities for pedestrians aged 56 and above in both blocks. (Hefang Street = 5.714, Gongchen Bridge = 8.000). For example, pedestrians stay longer in daily necessities store in Gongchen Bridge (1.250) than those in Hefang Street (0.476).

3.  The business staying time data about walkers of different types show (Table 5):

**Table 5.** Mean and standard deviation of dwell time in street furniture for different pedestrian types.

| | Utilitarian Walkers | | | | Recreational Walkers | | | | No-Specified Walkers | | | |
|---|---|---|---|---|---|---|---|---|---|---|---|---|
| | Hefang Street | | Gongchenqiao Street | | Hefang Street | | Gongchenqiao Street | | Hefang Street | | Gongchenqiao Street | |
| | N = 24 | | N = 38 | | N = 65 | | N = 83 | | N = 61 | | N = 29 | |
| | Mean | SD | Mean | SD | Mean | SD | Mean | SD | Mean | SD | Mean | SD |
| Pedestrian behavior (stay) | | | | | | | | | | | | |
| Essential facilities | | | | | | | | | | | | |
| Restaurant | 0.667 | 1.373 | 2.895 | 2.252 | 0.954 | 1.605 | 2.193 | 2.361 | 0.885 | 1.771 | 1.310 | 1.834 |
| Traditional Chinese Medicine pharmacy | 0.208 | 1.021 | 2.447 | 2.882 | 0.815 | 1.776 | 1.759 | 2.366 | 1.164 | 2.026 | 0.517 | 0.986 |
| Daily necessities store | 0.333 | 0.702 | 0.974 | 1.325 | 0.569 | 1.060 | 1.036 | 1.392 | 0.541 | 0.993 | 0.655 | 1.010 |
| Clothing | 0.750 | 1.511 | 1.053 | 1.723 | 1.138 | 1.609 | 1.807 | 1.797 | 0.770 | 1.244 | 1.414 | 1.268 |
| Commercial consumption facilities | | | | | | | | | | | | |
| Snack Bar | 1.042 | 1.160 | 1.395 | 1.480 | 1.492 | 1.301 | 1.506 | 1.173 | 1.115 | 1.318 | 1.379 | 1.208 |
| Milk tea beverage store | 0.500 | 0.885 | 0.737 | 1.131 | 0.569 | 1.104 | 1.277 | 1.417 | 0.738 | 1.237 | 1.034 | 1.180 |
| Dessert shop | 0.833 | 1.274 | 0.895 | 1.110 | 0.800 | 1.121 | 0.940 | 1.016 | 1.000 | 1.155 | 1.172 | 1.227 |
| Gift shop | 1.000 | 1.351 | 1.395 | 1.636 | 1.938 | 1.767 | 1.904 | 1.992 | 1.246 | 1.524 | 1.862 | 1.529 |
| Perfume shop | 0.375 | 1.013 | 0.368 | 0.883 | 0.431 | 0.770 | 0.940 | 1.253 | 0.639 | 1.001 | 0.690 | 0.806 |
| Specialty stores | 0.417 | 0.830 | 0.737 | 1.131 | 1.185 | 1.345 | 0.831 | 1.305 | 0.951 | 1.189 | 1.241 | 1.504 |
| Leisure and entertainment facilities | | | | | | | | | | | | |
| Web celebrity shop | 0.417 | 1.018 | 1.447 | 1.589 | 0.985 | 1.474 | 1.614 | 1.738 | 1.230 | 1.510 | 1.724 | 1.533 |
| Exhibition Hall | 0.958 | 2.010 | 2.158 | 2.707 | 2.385 | 2.415 | 3.277 | 2.760 | 1.475 | 2.142 | 2.414 | 2.428 |
| Public service facilities | | | | | | | | | | | | |
| Post card station | 0.792 | 1.414 | 1.184 | 1.449 | 0.738 | 1.492 | 1.795 | 1.924 | 1.033 | 1.653 | 1.862 | 1.747 |
| Park | 0.625 | 1.610 | 1.868 | 1.891 | 1.219 | 1.906 | 2.036 | 1.916 | 0.902 | 1.422 | 1.483 | 1.639 |
| Toilet | 0.292 | 0.624 | 0.711 | 0.898 | 0.508 | 0.937 | 0.940 | 1.052 | 0.525 | 0.849 | 0.931 | 0.923 |
| Rest seat | 0.917 | 1.472 | 1.737 | 2.140 | 1.985 | 1.663 | 2.398 | 2.252 | 1.590 | 2.077 | 2.276 | 1.888 |
| Sharable charger | 0.042 | 0.204 | 0.237 | 0.431 | 0.292 | 0.824 | 0.422 | 0.798 | 0.295 | 0.558 | 0.448 | 0.783 |

SD—standard deviation. Pedestrians (staying): "0" means "no staying"; "1" represents "within 1 min"; "2" means "1–5 min"; "3" represents "6–10 min"; "4" means "11–20 min"; "5" represents "20–30 min"; "6" means "over 30 min".

Utilitarian walkers (4.167) and recreational walkers (6.415) in Hefang Street stay in commercial consumption facilities for the longest time, while utilitarian walkers in Gongchen Bridge stay

in essential necessities (7.369) for the longest time and recreational walkers there stay in public service facilities (7.519) for the longest time. No-specified walkers stay in commercial consumption facilities (Hefang Street = 5.689, Gongchen Bridge = 7.378) for the longest time. The is the most obvious difference in no-specified walkers' staying time in public service facilities (Hefang Street = 4.345, Gongchen Bridge = 7) in both blocks. There is the most obvious difference between utilitarian walkers (HFJ = 3.476, GCQ = 6.795) and recreational walkers (Hefang Street = 3.476, Gongchen Bridge = 6.795) in terms of the staying time in essential facilities.

*4.4. Pedestrian Walking Experience*

(1)    The data of pedestrian walking experience of both genders demonstrate that (Table 6):

**Table 6.** Correlation analysis of daily walking time and multiple factors of pedestrians of different genders on two streets.

| Gender | The Overall | | | | Male | | | | Female | | | |
|---|---|---|---|---|---|---|---|---|---|---|---|---|
| | HFJ | t | GCQ | t | HFJ | t | GCQ | t | HFJ | t | GCQ | t |
| **Walking preference** | 0.352 | 0.000 *** | 0.274 | 0.001 *** | 0.227 | 0.137 | 0.294 | 0.031 ** | 0.340 | 0.008 *** | 0.173 | 0.114 |
| **Walking motivation** | | | | | | | | | | | | |
| Walking with children | −0.034 | 0.646 | 0.000 | 0.997 | −0.156 | 0.166 | −0.028 | 0.832 | 0.072 | 0.496 | 0.032 | 0.784 |
| Walking to school | 0.078 | 0.301 | 0.090 | 0.309 | −0.106 | 0.376 | −0.139 | 0.340 | 0.213 | 0.056 * | 0.246 | 0.032 ** |
| Walking to work | −0.079 | 0.262 | −0.067 | 0.431 | −0.032 | 0.764 | −0.341 | 0.018 ** | −0.016 | 0.880 | 0.099 | 0.366 |
| Walking to shop | 0.099 | 0.247 | 0.141 | 0.084 * | 0.040 | 0.766 | 0.224 | 0.084 * | 0.210 | 0.066 * | 0.116 | 0.296 |
| Recreational walking | 0.240 | 0.008 *** | 0.047 | 0.579 | 0.250 | 0.092 * | −0.210 | 0.111 | 0.193 | 0.123 | 0.338 | 0.004 *** |
| Sports-based walking | −0.043 | 0.601 | −0.008 | 0.920 | 0.081 | 0.572 | 0.008 | 0.952 | −0.063 | 0.568 | −0.163 | 0.153 |
| **Environmental characteristics** | | | | | | | | | | | | |
| Social quality | −0.029 | 0.746 | 0.014 | 0.864 | 0.026 | 0.850 | 0.003 | 0.984 | 0.011 | 0.937 | 0.031 | 0.792 |
| Street quality | 0.106 | 0.318 | −0.048 | 0.552 | 0.387 | 0.020 ** | 0.056 | 0.666 | −0.181 | 0.237 | 0.004 | 0.971 |
| Social safety | −0.034 | 0.646 | −0.024 | 0.769 | −0.156 | 0.166 | −0.157 | 0.287 | 0.072 | 0.496 | −0.009 | 0.932 |
| **LUM** | | | | | | | | | | | | |
| Essential facilities | 0.057 | 0.405 | 0.135 | 0.100 | 0.038 | 0.735 | 0.310 | 0.022 ** | 0.020 | 0.842 | −0.038 | 0.725 |
| Commercial consumption facilities | −0.110 | 0.126 | −0.213 | 0.063 * | −0.162 | 0.181 | −0.164 | 0.358 | −0.140 | 0.189 | −0.279 | 0.074 * |
| Leisure and entertainment facilities | 0.102 | 0.194 | 0.165 | 0.079 * | 0.052 | 0.643 | 0.219 | 0.132 | −0.086 | 0.461 | 0.168 | 0.190 |
| Public service facilities | −0.064 | 0.444 | −0.115 | 0.303 | 0.047 | 0.714 | −0.011 | 0.953 | −0.216 | 0.075 * | −0.120 | 0.431 |

*** $p < 0.01$, ** $p < 0.05$, * $p < 0.1$.

Generally, walking preferences (Hefang Street: r = 0.352, $p < 0.01$, Gongchen Bridge: R = r = 0.274, $p < 0.01$) significantly impact pedestrian walking experience in a positive manner in both blocks. In terms of walking motivation, recreational walking (r = 0.240, $p < 0.01$) impacts pedestrian walking experience in a positive manner in Hefang Street. Walking to shop (r = 0.141, $p < 0.1$) significantly impacts pedestrian walking experience in a positive manner in Gongchen Bridge. In terms of environmental characteristics, the distribution of commercial consumption facilities (r = −0.213, $p < 0.1$) significantly impacts the pedestrian walking experience in a negative manner in Gongchen Bridge. While the distribution of leisure and entertainment facilities (r = 0.165, $p < 0.1$) is negatively correlated with pedestrian walking experience.

Walking preferences (r = 0.294, $p < 0.05$) significantly impact the walking experience of male pedestrians in a positive manner in Gongchen Bridge. In terms of walking motivation, recreational walking (r = 0.250, $p < 0.1$) greatly impacts the walking experience of male pedestrians in a positive manner in Hefang Street. Walking to work (r = −0.341, $p < 0.05$)

and walking to shop (r = 0.224, $p < 0.1$) greatly influence the walking experience of male pedestrians in Gongchen Bridge. The former is negatively correlated with the walking experience, while the latter is positively correlated with it. In terms of environmental characteristics, street quality (r = 0.387, $p < 0.05$) significantly impacts the walking experience of male pedestrians in a positive manner in Hefang Street. The distribution of essential facilities (r = 0.310, $p < 0.05$) has a significant impact on the walking experience of male pedestrians in a positive manner in Gongchen Bridge.

Walking preference (r = 0.340, $p < 0.01$) has a significant impact on the walking experience of female pedestrians in a positive manner in Hefang Street. In terms of walking motivation, walking to school (r = 0.213, $p < 0.1$) and walking to shop (r = 0.210, $p < 0.1$) have significant influence on the walking experience of female pedestrians in a positive manner in Hefang Street. Walking to school (r = 0.246, $p < 0.05$) and recreational walking (r = 0.338, $p < 0.01$) have significant influence on the walking experience of female pedestrians in a positive manner in Gongchen Bridge. In terms of environmental characteristics, the distribution of public service facilities (r = −0.216, $p < 0.1$) significantly impacts the walking experience of female pedestrians in a negative manner in Hefang Street. The distribution of commercial consumption facilities (r = −0.279, $p < 0.1$) significantly impacts the walking experience of female pedestrians in a negative manner in Gongchen Bridge.

(2) The data of pedestrian walking experience of different age groups show that (Table 7):

**Table 7.** Correlation analysis of daily walking time and multiple factors of pedestrians of different ages on two streets.

| Age | 1–30 | | | | 31–55 | | | | 56 or Above | | | |
| --- | --- | --- | --- | --- | --- | --- | --- | --- | --- | --- | --- | --- |
| | HFJ | t | GCQ | t | HFJ | t | GCQ | t | HFJ | t | GCQ | t |
| **Walking preference** | 0.389 | 0.003 *** | 0.355 | 0.065 * | 0.218 | 0.256 | 0.196 | 0.096 * | −0.283 | 0.443 | 0.300 | 0.290 |
| **Walking motivation** | | | | | | | | | | | | |
| Walking with children | −0.035 | 0.718 | −0.033 | 0.902 | 0.017 | 0.899 | −0.089 | 0.462 | 0.212 | 0.386 | −0.063 | 0.928 |
| Walking to school | 0.144 | 0.222 | 0.066 | 0.799 | −0.092 | 0.551 | 0.067 | 0.562 | −0.154 | 0.665 | 0.611 | 0.248 |
| Walking to work | 0.012 | 0.913 | 0.006 | 0.980 | −0.445 | 0.006 *** | −0.054 | 0.637 | 0.082 | 0.804 | −0.050 | 0.888 |
| Walking to shop | 0.206 | 0.097 * | 0.100 | 0.616 | 0.264 | 0.214 | **0.186** | 0.128 | **0.346** | 0.292 | 0.516 | 0.201 |
| Recreational walking | 0.221 | 0.115 | 0.084 | 0.646 | 0.405 | 0.015 ** | −0.090 | 0.464 | 0.007 | 0.986 | 0.111 | 0.841 |
| Sports-based walking | 0.060 | 0.605 | 0.136 | 0.479 | −0.066 | 0.694 | −0.123 | 0.319 | −0.602 | 0.118 | **0.785** | 0.222 |
| **Environmental characteristics** | | | | | | | | | | | | |
| Social quality | −0.217 | 0.132 | **−0.268** | 0.212 | −0.063 | 0.659 | **0.158** | 0.227 | **1.042** | 0.093 * | 0.687 | 0.345 |
| Street quality | **0.234** | 0.130 | −0.167 | 0.402 | **0.181** | 0.043 ** | 0.015 | 0096 * | 0.212 | 0.186 | **0.823** | 0.094 * |
| Social safety LUM | −0.035 | 0.718 | −0.167 | 0.335 | 0.017 | 0.899 | 0.044 | 0.708 | −0.154 | 0.665 | −0.261 | 0.503 |
| Essential facilities | 0.064 | 0.504 | −0.012 | 0.942 | −0.053 | 0.722 | 0.140 | 0.241 | −0.110 | 0.663 | 0.387 | 0.264 |
| Commercial consumption facilities | −0.084 | 0.382 | **−0.203** | 0.415 | −0.089 | 0.647 | **−0.158** | 0.352 | 0.310 | 0.204 | 0.068 | 0.927 |
| Leisure and entertainment facilities | −0.044 | 0.662 | −0.055 | 0.791 | **0.227** | 0.126 | 0.072 | 0.574 | 0.113 | 0.787 | −0.009 | 0.990 |
| Public service facilities | **0.100** | 0.423 | −0.043 | 0.875 | −0.115 | 0.530 | −0.022 | 0.884 | **−0.528** | 0.033 ** | **−0.786** | 0.250 |

*** $p < 0.01$, ** $p < 0.05$, * $p < 0.1$.

Pedestrian preference of pedestrians aged 1–30 in Hefang Street and Gongchen Bridge (Hefang Street: r = 0.389, $p < 0.01$, Gongchen Bridge: r = 0.355, $p < 0.1$) is positively correlated with the total walking time. The correlation between pedestrian preference and total walking time in Hefang Street is stronger than that in Gongchen Bridge. It indicates that the walking

experience of pedestrians aged 1–30 in business block is more likely to be influenced by walking preference. The walking motivation of pedestrians aged 1–30 in Hefang Street (r = 0.206, $p < 0.1$) impacts their walking experience the most, while that of pedestrians aged 1–30 in Gongchen Bridge has no significant influence on their walking experience. The environmental characteristics have no significant impact on pedestrians aged 1–30 in both blocks.

In addition, walking preference has no significant impact on pedestrians aged 31–55 in both blocks. The walking motivation of walking to work for pedestrians aged 31–55 (r = −0.445, $p < 0.01$) in Hefang Street is negatively correlated with their walking experience, while the walking motivation of recreational walking (r = 0.405, $p < 0.05$) for those aged 31–55 in Hefang Street is positively correlated with their walking experience. The walking motivation has no obvious impact on walking experience for those aged 31–55 in Gongchen Bridge. Social quality in the environmental characteristics (Hefang Street: r = 0.181, $p < 0.05$, Gongchen Bridge: r = 0.015, $p < 0.1$) significantly impacts the walking experience of pedestrians aged 31–55 in a positive manner in both blocks.

Besides, walking preference and walking motivation have no significant impact on pedestrians aged 56 and above in both blocks. The social quality in the environmental characteristics (r = 0.823, $p < 0.1$) has a significant impact on the walking experience of pedestrians aged 56 and above in a positive manner in Gongchen Bridge. LUM of public service facilities (r = −0.528, $p < 0.05$) is negatively correlated with the walking experience of pedestrians aged 56 and above in Gongchen Bridge.

(3) The data of walking experience of walkers of different types show that (Table 8):

**Table 8.** Correlation analysis of daily walking time and multiple factors of different walker types on two streets.

| | Utilitarian Walkers | | | | Recreational Walkers | | | | No-Specified Walkers | | | |
|---|---|---|---|---|---|---|---|---|---|---|---|---|
| | **HFJ** | **t** | **GCQ** | **t** | **HFJ** | **t** | **GCQ** | **t** | **HFJ** | **t** | **GCQ** | **t** |
| **Walking preference** | 0.772 | 0.129 | 0.104 | 0.502 | 0.443 | 0.001 *** | 0.290 | 0.013 ** | 0.264 | 0.152 | 0.636 | 0.036 ** |
| **Walking motivation** | | | | | | | | | | | | |
| Walking with children | −0.248 | 0.462 | 0.225 | 0.190 | −0.108 | 0.347 | −0.120 | 0.314 | −0.189 | 0.230 | 0.068 | 0.808 |
| Walking to school | −0.383 | 0.215 | 0.394 | 0.039 ** | −0.002 | 0.984 | 0.122 | 0.341 | 0.157 | 0.274 | 0.136 | 0.673 |
| Walking to work | −0.907 | 0.158 | −0.045 | 0.776 | 0.001 | 0.990 | −0.092 | 0.437 | −0.126 | 0.332 | −0.180 | 0.483 |
| Walking to shop | 0.862 | 0.139 | −0.192 | 0.227 | −0.026 | 0.827 | 0.271 | 0.026 ** | 0.122 | 0.496 | 0.188 | 0.470 |
| Recreational walking | 0.197 | 0.647 | 0.296 | 0.172 | −0.011 | 0.930 | −0.013 | 0.913 | 0.345 | 0.057 * | 0.003 | 0.991 |
| Sports-based walking | 0.396 | 0.321 | −0.329 | 0.121 | −0.207 | 0.060 * | 0.009 | 0.942 | −0.058 | 0.749 | 0.207 | 0.480 |
| **Environmental characteristics** | | | | | | | | | | | | |
| Social quality | −0.362 | 0.292 | 0.162 | 0.398 | 0.007 | 0.955 | −0.063 | 0.594 | 0.071 | 0.678 | −0.130 | 0.703 |
| Street quality | 0.502 | 0.277 | −0.235 | 0.181 | 0.121 | 0.537 | 0.039 | 0.731 | −0.005 | 0.977 | 0.075 | 0.785 |
| Social safety | −0.248 | 0.462 | 0.025 | 0.871 | −0.108 | 0.347 | −0.095 | 0.422 | −0.189 | 0.230 | 0.385 | 0.237 |
| **LUM** | | | | | | | | | | | | |
| Essential facilities | 0.093 | 0.835 | 0.657 | 0.001 *** | 0.060 | 0.581 | −0.036 | 0.745 | 0.076 | 0.530 | 0.211 | 0.436 |
| Commercial consumption facilities | 0.285 | 0.461 | −0.114 | 0.560 | −0.021 | 0.847 | −0.199 | 0.171 | −0.184 | 0.240 | −0.349 | 0.495 |
| Leisure and entertainment facilities | 0.174 | 0.640 | 0.454 | 0.016 ** | −0.015 | 0.895 | 0.083 | 0.540 | 0.010 | 0.949 | 0.177 | 0.563 |
| Public service facilities | 0.936 | 0.129 | −0.412 | 0.128 | −0.353 | 0.008 *** | −0.176 | 0.193 | −0.062 | 0.690 | −0.077 | 0.867 |

*** $p < 0.01$, ** $p < 0.05$, * $p < 0.1$.

Walking preference (Hefang Street: r = 0.443, $p < 0.01$, Gongchen Bridge: r = 0.037, $p < 0.05$) significantly impacts the walking experience of recreational walkers in a positive manner.

Additionally, the walking motivation and environmental characteristics have no significant influence on the walking experience of utilitarian walkers in Hefang Street but have great influence on the walking experience of utilitarian walkers in Gongchen Bridge. Here, walking to school (r = 0.394, *p* < 0.05), street safety (r = 0.025, *p* < 0.1), distribution of essential necessities (r = 0.657, *p* < 0.01) and leisure and entertainment facilities (r = 0.454, *p* < 0.05) are positively correlated with the walking experience of utilitarian walkers.

In addition, walking motivation and environmental characteristics significantly impact the walking experience of recreational walkers in both blocks. The street quality (Hefang Street: r = 0.121, *p* < 0.1, Gongchen Bridge: r = 0.037, *p* < 0.1) is positively correlated with the walking experience of recreational walkers in both blocks. Sports-based walking (r = −0.207, *p* < 0.1) and the distribution of public service facilities (r = −0.353, *p* < 0.01) are negatively correlated with the walking experience of recreational walkers in Hefang Street. Walking to shop (r = 0.271, *p* < 0.05) and street quality (r = 0.039, *p* < 0.05) are positively correlated with the walking experience of recreational walkers in Gongchen Bridge.

Moreover, environmental characteristics have no significant impact on the walking experience of no-specified walkers in Hefang Street, while the walking motivation has a significant impact on the walking experience of no-specified walkers in Hefang Street. Recreational walking (r = 0.345, *p* < 0.1) is positively correlated with the walking experience of no-specified pedestrians in Hefang Street. It is worth noting that walking motivation and environmental characteristics have no significant influence on the walking experience of utilitarian walkers in Gongchen Bridge.

## 5. Discussion

To find specific factors impacting pedestrian walking behavior, create a favorable environment in the historical and cultural block, and benefit the pedestrian walking experience in the street, an objective approach is adopted in this paper to analyze pedestrian behavior in the street and demonstrate the correlation between (pedestrians' own factors and the street environment) and (pedestrian behavior and pedestrian walking experience). As for study methods, questionnaires and Statistical Product and Service Solutions (SPSS) data analysis are used in the study to more objectively and effectively decompose the correlations between various factors. In terms of the study content, the attention was paid to the walking of dynamic level in recent study in recent years. While this paper focuses more on the walking behavior of pedestrians who are lingering, and a relevant study is therefore conducted. Meanwhile, in previous studies, more attention was paid to the environmental construction and traffic planning of physical environment characteristics, and less attention was paid to the perspectives of the street cultural atmosphere and historical appearance. From the perspective of pedestrian walking behavior and walking experience, this paper explores the comprehensive characteristics of pedestrian walking in cultural heritage streets qualitatively and quantitatively. Multi-angle research can more accurately grasp the factors affecting pedestrian walking, provide empirical evidence for better building street space environment, provide research data support for the renovation and renewal of heritage streets in the future, and promote the related research on the improvement of street walking environment to a certain extent.

Based on the cultural attributes of historical and cultural blocks, the historical evolution, spatial scale and business distribution of the two blocks in the article are studied. Then the differences of pedestrian walking behaviors and walking experiences of different age groups are specifically analyzed. It's confirmed that the walking preference is generally positively correlated with the total walking time, with the influencing factor of pedestrian walking preference in Hefang Street business block being 0.352, and that of Gongchen Bridge life block being 0.274. In addition, due to the difference of pedestrians' walking motivation and block environment characteristics, and the difference in pedestrian's walking time in these two blocks, there are different correlations between LUM and walking time. The influence coefficient of social quality dimension in Hefang Street is −0.029, while that in Gongchen Bridge is (+)0.014. Meanwhile the influence coefficient of street quality dimension

in Hefang Street is 0.106, but that in Gongchen Bridge is −0.048. Above all, environmental characteristics have totally different influences on walking time in different blocks.

*5.1. Differences of Pedestrian Walking Behaviors*

It's found from the multiple linear regression analysis of both genders, daily walking time and multiple factors in Table 6, that for different genders of pedestrians, different pedestrian walking experiences will be produced.

(1) The correlation between gender and walking behaviors

Pedestrians' staying time in Gongchen Bridge life street is longer than that in Hefang Street business block. Pedestrians around the business block prefer staying in commercial consumption facilities, while pedestrians around the life block prefer staying in public service facilities. Male pedestrians prefer staying in commercial consumption facilities in business block, while they prefer staying in public service facilities in life block. Female pedestrians prefer staying in commercial consumption facilities in both blocks, which was possibly triggered by marginal effects. Previous studies have demonstrated that travelling time significantly impacts walking behaviors of both genders. Distance significantly impacts pedestrians' walking tendency. The marginal effect of distance is greater among women than among men in work trips, while the opposite is true in shopping trips [40];

(2) The relationship between different age groups and walking behaviors.

Commercial consumption facilities have a great influence on the walking behaviors of pedestrians in both blocks. It is worth noting that commercial consumption facilities have a great influence on the walking behaviors of pedestrians over 30 years old in business block, and public service facilities have a great influence on the walking behaviors of pedestrians over 30 years old in life block. It is shown in a study that young consumers show increasing spending power, and their spending rate is much higher than that of previous generations [41]. Compared with the elderly consumers, young people have the highest level of income, savings, and expenditure. Consumers over 65 years old show stronger emotional brand attachment than those aged 50 to 65 [42]. At present, the newly established business model is more likely to attract pedestrians aged 1–30;

(3) The relationship between walkers of different types and walking behaviors.

In both blocks, commercial consumption facilities have a strong influence on the pedestrian walking behaviors of different types. Existing studies have shown that blocks with various shopping options will attract pedestrians to stay for a long time, and shopping activities in blocks will positively impact pedestrians [43]. In addition, the essential facilities have a strong influence on the walking behaviors of utilitarian pedestrians in life block, while the public service facilities have an influence on the walking behaviors of recreational walkers and no-specified walkers.

*5.2. Differences in Pedestrian Walking Experience*

(1) The relationship between both genders and walking behaviors

Male pedestrians' walking experience in life block is influenced by their walking preferences. In business block, the walking motivation that has a great influence on male pedestrians' walking experience is recreational walking. In terms of environmental characteristics, street quality impacts their walking experience. For example, street landscape, architectural style, street cleanliness and connectivity are likely to influence their walking experience. Walking motivations that have great influence on male pedestrians' walking experience in life block involve walking to work and walking to shop. The influence of environmental characteristics on their walking experience can be reflected in the distribution of essential facilities. That's possibly because that most male pedestrians in Hefang Street are students who go for entertainment. The existing research shows that the student group has evolved into the main consumption force [44], and the walking environment impacts pedestrians' sense of security. While males who like walking boast a higher sense

of security [45], and the marginal effect of males' shopping distance is greater than that of females' [40]. Female pedestrians' walking experience in business block is influenced by their walking preference. Walking motivations in business block that have great influence on female pedestrians' walking experience include walking to school and walking to shop. The influence of environmental characteristics on their walking experience is reflected in the distribution of public service facilities. Walking motivations that have great influence on female pedestrians' walking experience in life block are walking to school and recreational walking. The influence of environmental characteristics on their walking experience is reflected in the distribution of commercial consumption facilities. Previous studies have shown that women's extroversion and openness to experience give them a stronger desire to buy compared with men [46], while there is not much commercial development in Gongchen Bridge life block, and most female walkers are college students, mainly travelling with friends. Close social distance promotes pedestrians' consumption behaviors [47]. While in Hefang Street business block with concentrated businesses, the consumption purpose is very clear. While the scattered businesses of Gongchen Bridge make pedestrians more likely to wander;

(2)　Relationship between different age groups and walking experience

Previous studies have shown that younger teenagers tend to be more physically active and prefer walking [48]. The data of this study show that the walking preference of pedestrians aged 1–30 has a greater impact on their total walking time. Most pedestrians aged 1–30 in Hefang Street and Gongchen Bridge, attend elementary school or junior high school. They hang out in the pedestrian block together with their family members. In the above research, social quality is positively correlated with the overall walking time of pedestrians aged 31–55 in Hefang Street and pedestrians aged 31–55, 56 and above in Gongchen Bridge. Previous studies have shown that the relationship between built environment and elderly people's travelling behaviors can be explained by peer effect or collective socialization [49]. The block activity richness and social interaction have a strong influence on middle-aged and elderly people. Neighborhood is an outdoor space, a transportation place, and an important area of the elderly friendly community. Studies have shown that there is a positive correlation between walking activities and social interaction [50]. This view is verified in our research. For the elderly, the harm caused by their lack of spiritual comfort is even more serious than that caused by physical diseases. With the acceleration of population aging, the geographical relationship between neighbors and friends in public space is particularly important for the mental health of the elderly;

(3)　Relationship between walkers of different types and walking experience.

The walking experience of recreational walkers will be strongly influenced by their walking preferences. Additionally, utilitarian walkers' walking experience is strongly influenced by street safety, the distribution of essential facilities: restaurants, Traditional Chinese Medicine pharmacies, daily necessities store and clothing stores, as well as the number of leisure and entertainment facilities: web celebrity shops and exhibition halls. Studies have shown that utilitarian walkers, as the name suggests, are utilitarian [51]. They walk mainly for commuting but pay little attention to social quality and street quality. Some studies have also shown that there is a causal relationship between walking experience and street safety values [52]. In addition, the walking experience of recreational walkers is strongly influenced by street quality referring to street landscape style, street architectural style, street cleanliness and street connectivity. The research shows that the key factor impacting the evaluation of recreational walking comfort is street quality [53]. The construction of infrastructures such as green road has a positive impact on the increase of pedestrians' weekly walking time [54]. The walking experience of recreational walkers in business block will be impacted by the distribution and quantity of public service facilities, such as postcard stations, parks, toilets, rest seats and sharable chargers. Recreational walking is a factor that significantly influences the walking experience of non-specified

pedestrians in business block. Previous studies have shown that recreational walking is more likely to produce positive and healthy emotions than utilitarian walking [55];

(4)   Comparison of existing studies

The study result here is similar to that from researchers who study on pedestrians' walking behavior and walking experience. The block business distribution tends to have a greater impact on street dynamics than the block cultural characteristics. Compared with other environment characteristics, the dimension of functional feature nurtures street dynamics. The business reflects pedestrians' immediate needs. It's found the study that higher LUM facilitates commuting and leisure walking, instead of walking of practical nature [56]. In this study, a variability exists in the effect of LUM on pedestrian walking behavior for streets of different types. In Hefang Street block of greater walkability, the utility of walking to work and walking to study is not significantly impacted by LUM, and the leisure walking coefficient of walking to shop sees a significant rise. While a different result is demonstrated in the life-oriented Gongchen Bridge block, where the utility of walking to school is significantly impacted by LUM.

## 6. Conclusions and Implications

### 6.1. Conclusions

The main conclusions of this study can be summarized as follows:

(1)   Pedestrian walking behavior

The walking behavior of those aged 1–30 is greatly influenced by commercial consumption facilities. The walking behavior of pedestrians aged over 30 in the business block is greatly influenced by the commercial consumption facilities, and the walking behavior of those aged over 30 in the life block is greatly influenced by the public service facilities. In both blocks, recreational walkers, utilitarian walkers, and no-specified walkers will stay in commercial consumption facilities for a long time. Essential facilities will have an impact on the walking behavior of pedestrians in life block, and public service facilities will have an impact on the walking behavior of pedestrians in business block;

(2)   Pedestrian experience

Men prefer walking in life block, while women prefer walking in business block. The street quality of business block is positively correlated with men's walking time. The street landscape, architectural style, cleanliness and connectivity have a significant impact on men's walking experience in business block. Men's walking experience in life block is greatly influenced by the distribution of essential facilities, while women's walking experience is greatly influenced by the distribution of public service facilities in business block. Walking preference has a strong influence on the walking experience of pedestrians aged 1–30, but a weak influence on the walking experience of pedestrians over 30. Street environmental characteristics, especially social quality, are positively correlated with the total walking time of pedestrians aged 56 and above. Activity richness and social interaction have a significant impact on the walking experience of pedestrians aged 56 and above. Walking preference will impact the walking experience of recreational walkers, and street quality will influence the walking experience of recreational walkers. Here, street landscape style, street architectural style, street cleanliness and street connectivity are important factors (in terms of street quality).

As the closest way to daily life and the healthiest way to environmental protection, it is necessary and meaningful to study the correlation between environment and pedestrian behavior, update and improve walking environment, stimulate pedestrian walking behavior, and enhance pedestrian walking experience. First, it can improve the health level of individuals and promote more pedestrians to participate in the cultural heritage blocks by improving the walking environment of cultural heritage streets, so that their body and mind can develop together. Second, the construction of neighborhood atmosphere and the improvement of quality of life create more opportunities for pedestrians to stay, build

street space with humanistic care and social atmosphere, enhance the overall vitality of cultural heritage streets, and improve the quality of life of pedestrians, which is also of great significance to public health.

*6.2. Implications*

The main recommendations of the study can be summarized as followed:

(1)    For pedestrian walking behaviors

As the commercial consumption facilities have a universal influence on the walking behaviors of pedestrians of all types, attention should be paid to the role of commercial consumption facilities in block. The consumption preferences of pedestrians of all age groups should be considered in the commercial consumption facilities of business block. But in both types of blocks, the setting of business types can be mainly aimed at pedestrians aged 1–30. The commercial consumption facilities inside the block should be considered. In addition, amid the construction of the block, the connection with the surrounding commercial complexes must be strengthened.

There is a strong correlation between the walking behavior of pedestrians over 30 years old and the block types. When the life block is designed, meeting the basic living needs of the surrounding residents should be considered, essential facilities should be diversified, so that residents can enjoy greater convenience. In addition, public service facilities should be fully improved in life block. When the block environment is renewed and constructed, the design that is friendly to the elderly can be prioritized. The full play is given to the historical and cultural functions of streets, so that the historical and cultural functions of street spaces can impact the physical and mental health of pedestrians, promote walking exercise, and improve public health;

(2)    For pedestrian walking experience

Pedestrians of different genders have different preferences for block types. When various types of blocks are planned and designed, attention should be paid to pedestrians of the target gender and the preferences of pedestrians of the other gender should be catered in other aspects. For example, if life blocks mainly attract male pedestrians, amid construction, the walking experience of male pedestrians in essential facilities and street quality must be mainly considered. Besides, more attention should be paid to the walking experience of female pedestrians in commercial consumption facilities and public service facilities, so as to meet the walking needs of pedestrians of both genders.

As the pedestrian's walking preference over 30 years old has a weak influence on the walking experience, the influence of other factors on pedestrians' walking experience should be fully considered in the process of block construction. For example, the needs of the elderly for social quality, activity richness and social interaction should be considered. Specifically, the circulation and landscape of the block can be improved. A platform for communication can be provided to strengthen the block interaction.

*6.3. Limitations and Future Research*

There is a higher correlation between the walking experience of recreational walkers and various influencing factors, so the opinions of such walkers are more likely to deserve attention. In addition, before the block is designed, the structures of pedestrians of different types should be investigated. Corresponding design strategies must be made according to the pedestrian structure after the site properties and pedestrian opinions are fully considered.

There are some limitations in the study. For example, pedestrians are picked up selectively rather than randomly to complete the questionnaires. Therefore, there may be some bias in the whole process of data statistics. Moreover, some respondents are impatient or pay less attention while filling out questionnaires. Thus, the subsequent analysis is possibly hampered by the lack of authenticity and reliability of the information.

Due to the difference in street attributes, or the variability in the streets where respondents are, the study conclusion is impacted to some degree. The pedestrians in the

article can't fully represent all populations due to the differences in personal circumstances, insufficient sample size, and uneven distribution of population of different information. Thereby, the validity and generalizability of the conclusions obtained in the article are possibly affected. In addition, direct observation is adopted, and the observation result is more likely to be influenced by subjective factors and the specific surrounding environments. The observer's attention and memory alone cannot make sure the complete record of pedestrians' complex behavioral activities. Also, the observation result is impacted by various factors.

It's found from the pedestrian information of the research samples, there is a small number of young and old respondents. There is not sufficient research on these two age groups, which then will be supplemented in subsequent studies. Besides, due to the investigation time, it is not possible to determine the impact of seasonal and climatic factors on pedestrian behaviors in the street space. It is also a key consideration for future research.

**Author Contributions:** In this paper, M.T. performed the conceptualization, Z.L. and Z.X. proposed the methodology, Q.X., Y.P. and T.C. performed the site investigation and data calculation, T.D. reviewed it and refined the diagram. All authors (Z.L., M.T., Q.X., Y.P., T.C., T.D. and Z.X.) organized the paper structure, wrote and edited it. All authors have read and agreed to the published version of the manuscript.

**Funding:** This research was funded by Humanities and Social Sciences Research Project of the Chinese Ministry of Education, grant number 18YJC760078; Zhejiang University of Technology Liberal Arts Laboratory, Database Construction Project "Zhejiang Historical and Cultural Villages and Towns Spatial Humanities Database", grant number GZ21681180011; Humanities and Social Sciences Fund of Zhejiang Education Department, grant number Y202146952.

**Institutional Review Board Statement:** Not applicable.

**Informed Consent Statement:** Not applicable.

**Data Availability Statement:** Not applicable.

**Acknowledgments:** The author gratefully acknowledges the editors and referees for their positive and constructive comments in the review process.

**Conflicts of Interest:** The authors declare that they have no known competing financial interests or personal relationships that could have appeared to influence the work reported in this paper.

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
