# Peer review of "Walking in China’s Historical and Cultural Streets: The Factors Affecting Pedestrian Walking Behavior and Walking Experience"

_land, doi:10.3390/land11091491_

Round 1
Reviewer 1 Report
About the paper with the title "Walking in the China’s historical and cultural streets: The factors affecting pedestrian walking behavior and walking experience" I suggest the following:
Improve the abstract highlighting better the objectives and methodologies.
Improve the methodologies explanation in the section 2 and 3 (maybe may be merged). For example, what is the dimension of the sample?
May be important to reanalyse the copyright of figures.
Maybe some subsections and points may be merged.
It could be important to present practical implications of the research and policy recommendations.
Author Response
To Editors and Reviewers
Dear Academic Editor,
Thank you for your letter and for the reviewers’ comments concerning our manuscript entitled “Walking in the China’s historical and cultural streets: The factors affecting pedestrian walking behavior and walking experience”.
Those comments are all valuable and helpful for revising and improving our paper, as well as the important guiding significance to our researches. We have studied comments carefully and have made correction. Revised portion are marked in red in the “land-revised-1878455”. The main corrections in the paper and the responds to the reviewer’s comments are as follows.
Responds to the editor’s comments:
1.It is necessary to add a paragraph outlining the physical landscape and its changes for to adapt the paper to the spirit of the S.I.
Answer:
Thank you very much for pointing out the problem. In the paper, we have added the physical landscape and its changes, and revised portion are marked in red in the “land-revised-1878455”. As following:
In 1999, the Hangzhou Municipal Government decided to redevelop the 13.6-hectare block adjacent to the Wushan(Town God’s Temple) square into an antique pedestrian street of commerce and tourism. The renovated Hefang Street was opened in October 2002, as shown in Figure 4. Nowadays, Hefang Street reveals the style of the late Qing Dynasty and early Ming Dynasty. The cultural value is highlighted here to create a marketplace culture of commerce, pharmaceutical industry and architecture. Its historical authenticity, cultural continuity and the overall appearance of the landscape are maintained. On both sides of Hefang Street, there are mainly local products, antiques, paintings and tourist souvenirs, and a craft pavilion is set up in the center of the street to display folk handicraft performances.
Now, there are many problems in Hefang Street. First of all, the public design at the monotonous node of the entrance space of Hefang Street is relatively simple. Most of them are seats combined with tree pools and simple humanoid sculptures. They are not attractive to walkers. The whole street is similar in format and has a high repetition rate. The feeling is bland and can not produce great fluctuations in the sensory psychology of walkers. The viewing speed of respondents has not changed significantly. Secondly, in the design process of Hefang Street, there is not enough leisure space at the door of the shop which is easy to arouse pedestrians' interest, and the influence of natural factors such as season and climate is not considered, which is easy to cause local space congestion due to natural factors such as sunshine and rain, which affects pedestrians' walking experience. Finally, due to the high degree of commercialization of blocks in the old city bring vitality, the excessive commercialization and destruction of the traditional charm of space in Hefang Street reflects the traditional culture. Except for the architectural appearance, most of them show the traditional life scene by sculpture. Such facilities can not interact with pedestrians, can not render the traditional cultural atmosphere of the street, can not leave a deep impression on pedestrians, and lack of interactive interest.
Responds to the reviewer’s 1 comments:
- Improve the abstract highlighting better the objectives and methodologies.
Answer:
Thank you very much for pointing out the problem. In the paper, we have improved the abstract, and revised portion are marked in the “land-revised-1878455”. As following:
Abstract: The urban street has evolved into an important indicator reflecting citizens’ living standard today, and pedestrian walking activity in the streets has been proved to be a major facilitator of public health. Uncertainties, however, exist in the factors affecting pedestrian walking behavior and walking experience in streets. Especially, the factors affecting pedestrian walking behavior and walking experience in the historical and cultural streets. For the study of their main influencing factors, Hefang Street business block and Gongchen Bridge life block in Hangzhou are selected here as the study objects. Both non-participatory and participatory research methods are adopted to collect pedestrian information and observe pedestrians’ ambiguous behavior, specific behavior, and stopping behavior. According to the study result, walking preference, walking time, environmental characteristics, and land-use mix (LUM) significantly impact pedestrian walking motivation. The type differences between Gongchen Bridge life block and Hefang Street business block leads to the difference in pedestrians' behaviors and their stopping time in business. Meanwhile, gender differences bring pedestrians’ significant differences in walking motivation. Pedestrian walking preference and walking time are positively correlated with walking motivation in both streets. Environmental characteristics and LUM have also been proved to be important influencing factors of pedestrians' walking motivation. In this article, design and planning strategies are proposed for streets of different types in an attempt to provide reference for the revitalization and utilization of cultural heritage streets.
- May be important to reanalyse the copyright of figures.
Answer:
Thank you very much for pointing out the problem. In the paper, we have revised the figures, and revised portion are marked in the “land-revised-1878455”.
- Maybe some subsections and points may be merged.
Answer:
Thank you very much for pointing out the problem. In the paper, we have merged some subsections, and revised portion are marked in the “land-revised-1878455”.
- It could be important to present practical implications of the research and policy recommendations.
Answer:
Thank you very much for pointing out the problem. In the paper, we have added the research and policy recommendations, and revised portion are marked in the “land-revised-1878455”. As following:
As the closest way to daily life and the healthiest way to environmental protection, it is necessary and meaningful to study the correlation between environment and pedestrian behavior, update and improve walking environment, stimulate pedestrian walking behavior and enhance pedestrian walking experience. First, it can improve the health level of individuals and promote more pedestrians to participate in the cultural heritage blocks by improving the walking environment of cultural heritage streets, so that their body and mind can develop together. Second, the construction of neighborhood atmosphere and the improvement of quality of life create more opportunities for pedestrians to stay, build street space with humanistic care and social atmosphere, enhance the overall vitality of cultural heritage streets and improve the quality of life of pedestrians, which is also of great significance to public health.
6.2. Implications
The main recommendations of the study can be summarized as followed:
(1)For pedestrian walking behaviors
As the commercial consumption facilities have a universal influence on the walking behaviors of pedestrians of all types, attention should be paid to the role of commercial consumption facilities in block. The consumption preferences of pedestrians of all age groups should be considered in the commercial consumption facilities of business block. But in both types of blocks, the setting of business types can be mainly aimed at pedestrians aged 1-30. The commercial consumption facilities inside the block should be considered. In addition, amid the construction of the block, the connection with the surrounding commercial complexes must be strengthened.
There is a strong correlation between the walking behavior of pedestrians over 30 years old and the block types. When the life block is designed, meeting the basic living needs of the surrounding residents should be considered, essential facilities should be diversified, so that residents can enjoy greater convenience. In addition, public service facilities should be fully improved in life block. When the block environment is renewed and constructed, the design that is friendly to the elderly can be prioritized. The full play is given to the historical and cultural functions of streets, so that the historical and cultural functions of street spaces can impact the physical and mental health of pedestrians, promote walking exercise, and improve public health.
(2)For pedestrian walking experience
Pedestrians of different genders have different preferences for block types. When various types of blocks are planned and designed, attention should be paid to pedestrians of the target gender and the preferences of pedestrians of the other gender should be catered in other aspects. For example, if life blocks mainly attract male pedestrians, amid construction, the walking experience of male pedestrians in essential facilities and street quality must be mainly considered. Besides, more attention should be paid to the walking experience of female pedestrians in commercial consumption facilities and public service facilities, so as to meet the walking needs of pedestrians of both genders.
As the pedestrian's walking preference over 30 years old has a weak influence on the walking experience, the influence of other factors on pedestrians’ walking experience should be fully considered in the process of block construction. For example, the needs of the elderly for social quality, activity richness and social interaction should be considered. Specifically, the circulation and landscape of the block can be improved. A platform for communication can be provided to strengthen the block interaction.
Responds to the reviewer’s 2 comments:
- The sentence in lines 218 and 219 sounds strange, not easy to understand, so it should be revised, as should the first sentence of the conclusion.
Answer:
Thank you very much for pointing out the problem. In the paper, we have revised the sentence, and revised portion are marked in the “land-revised-1878455”.
- Make references to photos, perspectives or plans as soon as they are mentioned in the text.
Answer:
Thank you very much for pointing out the problem. In the paper, we have added the references, and revised portion are marked in the “land-revised-1878455”.
- Titles 4.1 and 4.11 are the same, they should be differentiated.
Answer:
Thank you very much for pointing out the problem. In the paper, we have revised the titles, and revised portion are marked in the “land-revised-1878455”. As following:
4.1. Historical evolution
4.1.1. The historical evolution of Hefang Street and Gongchen Bridge life block
- Maps do not contain scales or north! Figure 5 is not clear at all, where is the Gongchen bridge? It seems that the map changes scale but it is not indicated anywhere. I recommend to show in grey the streets which are not the object of the study but which exist in its surroundings, thus allowing to better understand the global configuration of the road system surrounding the bridge. Otherwise, one could think that there are no streets in the vicinity of the study area, which is not true. Between figures 5 and 6 it seems that the areas and boundaries of the two study areas are not the same. Figure 6 seems to be more accurate because it is at a finer scale, but as this is not indicated it is not known whether the scale of representation is the same for the two study areas. As the amenities are then related to the total surface area of the study areas it is important to be able to compare the two, and therefore to see them represented at the same scale with a precise delimitation. Figures 3 and 6 need to be made legible, too small and/or poor definition and/or not in English.
Answer:
Thank you very much for pointing out the problem. In the paper, we have revised the maps, and figures, and revised portion are marked in the “land-revised-1878455”. As following:
Figure 2. Location map of Hefang Street business block and Gongchen Bridge life block.(according to Baidu map)
Figure 3. Bazaar paintings in the past (left), Street pattern map of southern Song Dynasty (middle) and Street pattern map of Ming and Qing Dynasties (right). (online website https://jz.docin.com/p-2811341437.html@)
Figure 4. Hefang Street in 1966 (left) and the current He Fang Street (right). (online website http://www.360doc.com/content/20/0812/10/17132703_929812656.shtml)
Figure 5. Morphological evolution of Gongchen Bridge Block. (Self-painted according to the historical evolution of Gongchen Bridge)
Figure 6: Site distribution of Hefang Street business block and Gongchen Bridge life block (according to Baidu map)
- Is there a metric somewhere on the distances travelled? On the point of departure/place of residence? On the duration of the travel time? On its relative share compared to other means of transport (bicycle, metro, cars, in the broader sense of intermodality...)
Answer:
Thank you very much for pointing out the problem. In the paper, the uniform unit of measurement in distance is meters. Questionnaire survey and tracking of pedestrians are completed in two historical streets. Entertainment walking is a classification of walking types, including pedestrians traveling in historical streets. In addition, the paper studies the influence factors of pedestrians' walking experience on historical streets without considering other means of transportation. In the future research, we will consider other means of transport. We have revised the content, and revised portion are marked in the “land-revised-1878455”.
- Be sure to systematically use the units of measurement for the figures presented, both in the text and in the table.
Answer:
Thank you very much for pointing out the problem. In the paper, we have presented the same units in the text and in the table, and revised portion are marked in the “land-revised-1878455”.
- The conclusion seems very long, I suggest that some parts could be added to the discussion.
Answer:
Thank you very much for pointing out the problem. In the paper, we have adjusted the content of conclusion and discussion, and revised portion are marked in the “land-revised-1878455”. As following:
- Discussion
To find specific factors impacting pedestrian walking behavior, create a favorable environment in the historical and cultural block, and benefit the pedestrian walking experience in the street, an objective approach is adopted in this paper to analyze pedestrian behavior in the street and demonstrate the correlation between (pedestrians’ own factors and the street environment) and (pedestrian behavior and pedestrian walking experience). As for study methods, questionnaires and Statistical Product and Service Solutions (SPSS) data analysis are used in the study to more objectively and effectively decompose the correlations between various factors. In terms of the study content, the attention was paid to the walking of dynamic level in recent study in recent years. While this paper focuses more on the walking behavior of pedestrians who are lingering and a relevant study is therefore conducted. Meanwhile, in previous studies, more attention was paid to the environmental construction and traffic planning of physical environment characteristics, and less attention was paid to the perspectives of the street cultural atmosphere and historical appearance. From the perspective of pedestrian walking behavior and walking experience, this paper explores the comprehensive characteristics of pedestrian walking in cultural heritage streets qualitatively and quantitatively. Multi-angle research can more accurately grasp the factors affecting pedestrian walking, provide empirical evidence for better building street space environment, provide research data support for the renovation and renewal of heritage streets in the future, and promote the related research on the improvement of street walking environment to a certain extent.
Based on the cultural attributes of historical and cultural blocks, the historical evolution, spatial scale and business distribution of the two blocks in the article are studied. Then the differences of pedestrian walking behaviors and walking experiences of different age groups are specifically analyzed. It’s confirmed that the walking preference is generally positively correlated with the total walking time, with the influencing factor of pedestrian walking preference in Hefang Street business block being 0.352, and that of Gongchen Bridge life block being 0.274. In addition, due to the difference of pedestrians’ walking motivation and block environment characteristics, and the difference in pedestrian’s walking time in these two blocks, there are different correlations between LUM and walking time. The influence coefficient of social quality dimension in Hefang Street is -0.029, while that in Gongchen Bridge is (+)0.014. Meanwhile the influence coefficient of street quality dimension in Hefang Street is 0.106, but that in Gongchen Bridge is -0.048. Above all, environmental characteristics have totally different influences on walking time in different blocks.
5.1. Differences of pedestrian walking behaviors
It’s found from the multiple linear regression analysis of both genders, daily walking time and multiple factors in Table 6, that for different genders of pedestrians, different pedestrian walking experiences will be produced.
(1) The correlation between gender and walking behaviors
Pedestrians’ staying time in Gongchen Bridge life street is longer than that in Hefang Street business block. Pedestrians around business block prefer staying in commercial consumption facilities, while pedestrians around life block prefer staying in public service facilities. Male pedestrians prefer staying in commercial consumption facilities in business block, while they prefer staying in public service facilities in life block. Female pedestrians prefer staying in commercial consumption facilities in both blocks. That possibly triggered by marginal effect. Previous studies have demonstrated that travelling time significantly impacts walking behaviors of both genders. Distance significantly impacts pedestrians’ walking tendency. The marginal effect of distance is greater among women than among men in work trips, while the opposite is true in shopping trips [40].
(2) The relationship between different age groups and walking behaviors.
Commercial consumption facilities have a great influence on the walking behaviors of pedestrians in both blocks. It is worth noting that commercial consumption facilities have a great influence on the walking behaviors of pedestrians over 30 years old in business block, and public service facilities have a great influence on the walking behaviors of pedestrians over 30 years old in life block. It’s shown in a study that young consumers show increasing spending power, and their spending rate is much higher than that of previous generations [41]. Compared with the elderly consumers, young people have the highest level of income, savings and expenditure. Consumers over 65 years old show stronger emotional brand attachment than those aged 50 to 65 [42]. At present, the newly established business model is more likely to attract pedestrians aged 1-30.
(3) The relationship between walkers of different types and walking behaviors.
In both blocks, commercial consumption facilities have a strong influence on the pedestrian walking behaviors of different types. Existing studies have shown that blocks with various shopping options will attract pedestrians to stay for a long time, and shopping activities in blocks will positively impact pedestrians [43]. In addition, the essential facilities have a strong influence on the walking behaviors of utilitarian pedestrians in life block, while the public service facilities have an influence on the walking behaviors of recreational walkers and no-specified walkers.
5.2. Differences in pedestrian walking experience
(1) The relationship between both genders and walking behaviors
Male pedestrians’ walking experience in life block is influenced by their walking preferences. In business block, the walking motivation that has a great influence on male pedestrians’ walking experience is recreational walking. In terms of environmental characteristics, street quality impacts their walking experience. For example street landscape, architectural style, street cleanliness and connectivity are likely to influence their walking experience. Walking motivations that have great influence on male pedestrians’ walking experience in life block involve walking to work and walking to shop. The influence of environmental characteristics on their walking experience can be reflected in the distribution of essential facilities. That’s possibly because that most male pedestrians in Hefang Street are students who go for entertainment. The existing research shows that the student group has evolved into the main consumption force [44], and the walking environment impacts pedestrians’ sense of security. While males who like walking boast a higher sense of security [45], and the marginal effect of male’s shopping distance is greater than that of female’ [46]. Female pedestrians’ walking experience in business block is influenced by their walking preference. Walking motivations in business block that have great influence on female pedestrians’ walking experience include walking to school and walking to shop. The influence of environmental characteristics on their walking experience is reflected in the distribution of public service facilities. Walking motivations that have great influence on female pedestrians’ walking experience in life block are walking to school and recreational walking. The influence of environmental characteristics on their walking experience is reflected in the distribution of commercial consumption facilities. Previous studies have shown that women’s extroversion and openness to experience give them a stronger desire to buy compared with men [47], while there is not much commercial development in Gongchen Bridge life block, and most female walkers are college students, mainly travelling with friends. Close social distance promotes pedestrians’ consumption behaviors [48]. While in Hefang Street business block with concentrated businesses, the consumption purpose is very clear. While the scattered businesses of Gongchen Bridge make pedestrians more likely to wander.
(2) Relationship between different age groups and walking experience
Previous studies have shown that younger teenagers tend to be more physically active and prefer walking [49]. The data of this study show that the walking preference of pedestrians aged 1-30 has a greater impact on their total walking time. Most pedestrians aged 1-30 in Hefang Street and Gongchen Bridge, attend elementary school or junior high school. They hang out in the pedestrian block together with their family members. In the above research, social quality is positively correlated with the overall walking time of pedestrians aged 31-55 in Hefang Street and pedestrians aged 31-55, 56 and above in Gongchen Bridge. Previous studies have shown that the relationship between built environment and elderly people’s travelling behaviors can be explained by peer effect or collective socialization [50]. The block activity richness and social interaction have a strong influence on middle-aged and elderly people. Neighborhood is an outdoor space, a transportation place, and an important area of the elderly friendly community. Studies have shown that there is a positive correlation between walking activities and social interaction [51]. This view is verified in our research. For the elderly, the harm caused by their lack of spiritual comfort is even more serious than that caused by physical diseases. With the acceleration of population aging, the geographical relationship between neighbors and friends in public space is particularly important for the mental health of the elderly.
(3) Relationship between walkers of different types and walking experience.
The walking experience of recreational walkers will be strongly influenced by their walking preferences. Additionally, utilitarian walkers’ walking experience is strongly influenced by street safety, the distribution of essential facilities: restaurants, Traditional Chinese Medicine pharmacies, daily necessities store and clothing stores, as well as the number of leisure and entertainment facilities: web celebrity shops and exhibition halls. Studies have shown that utilitarian walkers, as the name suggests, are utilitarian [52]. They walk mainly for commuting, but pay little attention to social quality and street quality. Some studies have also shown that there is a causal relationship between walking experience and street safety values [53]. In addition, the walking experience of recreational walkers is strongly influenced by street quality referring to street landscape style, street architectural style, street cleanliness and street connectivity. The research shows that the key factor impacting the evaluation of recreational walking comfort is street quality [54]. The construction of infrastructures such as green road has a positive impact on the increase of pedestrians’ weekly walking time [55]. The walking experience of recreational walkers in business block will be impacted by the distribution and quantity of public service facilities, such as postcard stations, parks, toilets, rest seats and sharable chargers. Recreational walking is a factor that significantly influences the walking experience of non-specified pedestrians in business block. Previous studies have shown that recreational walking is more likely to produce positive and healthy emotions than utilitarian walking [56].
(4)Comparison of existing studies
The study result here is similar to that from researchers who study on pedestrians’ walking behavior and walking experience. The block business distribution tends to have a greater impact on street dynamics than the block cultural characteristics. Compared with other environment characteristics, the dimension of functional feature nurtures street dynamics. The business reflects pedestrians’ immediate needs. It’s found the study that higher LUM facilitates commuting and leisure walking, instead of walking of practical nature [57]. In this study, a variability exists in the effect of LUM on pedestrian walking behavior for streets of different types. In Hefang Street block of greater walkability, the utility of walking to work and walking to study is not significantly impacted by LUM, and the leisure walking coefficient of walking to shop sees a significant rise. While a different result is demonstrated in the life-oriented Gongchen Bridge block, where the utility of walking to school is significantly impacted by LUM.
- Conclusion and implications
6.1. Conclusion
The main conclusions of this study can be summarized as follows:
(1)Pedestrian walking behavior
The walking behavior of those aged 1-30 is greatly influenced by commercial consumption facilities. The walking behavior of pedestrians aged over 30 in the business block is greatly influenced by the commercial consumption facilities, and the walking behavior of those aged over 30 in the life block is greatly influenced by the public service facilities. In both blocks, recreational walkers, utilitarian walkers and no-specified walkers will stay in commercial consumption facilities for a long time. Essential facilities will have an impact on the walking behavior of pedestrians in life block, and public service facilities will have an impact on the walking behavior of pedestrians in business block.
(2)Pedestrian experience
Men prefer walking in life block, while women prefer walking in business block. The street quality of business block is positively correlated with men’s walking time. The street landscape, architectural style, cleanliness and connectivity have a significant impact on men's walking experience in business block. Men's walking experience in life block is greatly influenced by the distribution of essential facilities, while women's walking experience is greatly influenced by the distribution of public service facilities in business block. Walking preference has a strong influence on the walking experience of pedestrians aged 1-30, but a weak influence on the walking experience of pedestrians over 30. Street environmental characteristics, especially social quality, are positively correlated with the total walking time of pedestrians aged 56 and above. Activity richness and social interaction have a significant impact on the walking experience of pedestrians aged 56 and above. Walking preference will impact the walking experience of recreational walkers, and street quality will influence the walking experience of recreational walkers. Here, street landscape style, street architectural style, street cleanliness and street connectivity are important factors (in terms of street quality).
As the closest way to daily life and the healthiest way to environmental protection, it is necessary and meaningful to study the correlation between environment and pedestrian behavior, update and improve walking environment, stimulate pedestrian walking behavior and enhance pedestrian walking experience. First, it can improve the health level of individuals and promote more pedestrians to participate in the cultural heritage blocks by improving the walking environment of cultural heritage streets, so that their body and mind can develop together. Second, the construction of neighborhood atmosphere and the improvement of quality of life create more opportunities for pedestrians to stay, build street space with humanistic care and social atmosphere, enhance the overall vitality of cultural heritage streets and improve the quality of life of pedestrians, which is also of great significance to public health.
- I found it difficult at times to understand what use the data produced by the article will be put to. It would be interesting to specify what a decision maker could do with it.
Answer:
Thank you very much for pointing out the problem. In the paper, we have revised the content, and revised portion are marked in the “land-revised-1878455”. As following:
The investigation is divided into two parts: field investigation and user investigation. The former consists of two parts: site investigation and behavior observation. Interview, photo-taking and other forms are adopted to obtain the distribution information of street businesses, including the distribution locations of businesses of different types and the proportion of the business of a certain type in the business as a whole. In terms of building scale and street paving and decoration, the building height, road width and paving material are investigated and analyzed. While behavior observation activity is performed, pedestrian behavior is observed and recorded in tracking investigation, that is, pedestrians are tracked in real time in the street and their status and staying time in shops are recorded. Different business types include essential necessities, commercial consumption facilities, leisure and entertainment facilities and public service facilities. In the user survey, pedestrians, staff of surrounding shops, recreational residents in the community and other street users are invited to fill out questionnaires for information collection. While some of pedestrians with poor educational background or the aged ones can’t read the questionnaires, so they are inquired orally and recorded. The questionnaire contents include pedestrian behaviors, socio-demographic characteristics, total walking time, walking preference, walking motivation and environmental characteristics. Information including the participants’ demographic data, their environment satisfaction and daily travelling behavior preferences are obtained in the questionnaire of the study.

Reviewer 2 Report
(1) Explain each acronym as it appears, why not add a glossary (LUM, SPSS,...)
(2) The sentence in lines 218 and 219 sounds strange, not easy to understand, so it should be revised, as should the first sentence of the conclusion
(3) Make references to photos, perspectives or plans as soon as they are mentioned in the text
(4) Titles 4.1 and 4.11 are the same, they should be differentiated
(5) Maps do not contain scales or north! Figure 5 is not clear at all, where is the Gongchen bridge? It seems that the map changes scale but it is not indicated anywhere. I recommend to show in grey the streets which are not the object of the study but which exist in its surroundings, thus allowing to better understand the global configuration of the road system surrounding the bridge. Otherwise, one could think that there are no streets in the vicinity of the study area, which is not true. Between figures 5 and 6 it seems that the areas and boundaries of the two study areas are not the same. Figure 6 seems to be more accurate because it is at a finer scale, but as this is not indicated it is not known whether the scale of representation is the same for the two study areas. As the amenities are then related to the total surface area of the study areas it is important to be able to compare the two, and therefore to see them represented at the same scale with a precise delimitation. Figures 3 and 6 need to be made legible, too small and/or poor definition and/or not in English.
(6) Is there a metric somewhere on the distances travelled? On the point of departure/place of residence? On the duration of the travel time? On its relative share compared to other means of transport (bicycle, metro, cars, in the broader sense of intermodality...)
(7) Be sure to systematically use the units of measurement for the figures presented, both in the text and in the table
(8) The conclusion seems very long, I suggest that some parts could be added to the discussion
(9) I found it difficult at times to understand what use the data produced by the article will be put to. It would be interesting to specify what a decision maker could do with it
Author Response

(The authors gave the same response as above.)
